# BACKDOORS STUCK AT THE FRONTDOOR: MULTI-AGENT BACKDOOR ATTACKS THAT BACKFIRE

## ABSTRACT

Malicious agents in collaborative learning and outsourced data collection threaten the training of clean models. Backdoor attacks, where an attacker poisons a model during training to successfully achieve targeted misclassification, are a major concern to train-time robustness. In this paper, we investigate a multi-agent backdoor attack scenario, where multiple attackers attempt to backdoor a victim model simultaneously. A consistent backfiring phenomenon is observed across a wide range of games, where agents suffer from a low collective attack success rate. We examine different modes of backdoor attack configurations, non-cooperation / cooperation, joint distribution shifts, and game setups to return an equilibrium attack success rate at the lower bound. The results motivate the re-evaluation of backdoor defense research for practical environments.

## 1 INTRODUCTION

Beyond training algorithms, the scale-up of model training depends strongly on the trust between agents. In collaborative learning and outsourced data collection training regimes, backdoor attacks and defenses (Gao et al., 2020; Li et al., 2021) are studied to mitigate a single malicious agent that perturbs train-time images for targeted test-time misclassifications. Outsourced data collection is common amongst industry practitioners, where Taigman et al. (2014); Papernot (2018) find a strong reliance on web-scraped data or third-party sourcing. Kumar et al. (2020) also find that data poisoning is viewed by practitioners as their most serious threat. In many practical situations, It is plausible in practice for >1 attacker, such as the poisoning of crowdsourced and agent-driven datasets on Google Images (hence afflicting subsequent scraped datasets) and financial market data respectively, or poisoning through human-in-the-loop learning on mobile devices or social network platforms. In this paper, we investigate the under-represented aspect of agent dynamics in backdoor attacks: what happens when *multiple* backdoor attackers are present? We simulate different configurations to study how the payoff landscape changes for attackers, with respect to standard attack/defense configurations, cooperative/non-cooperative behaviour, and joint distribution shifts. Our key contributions are:

- We explore the novel scenario of the multi-agent backdoor attack. Our findings on the backfiring effect and a low equilibrium attack success rate indicate a stable, natural defense against backdoor attacks, and motivates us to propose the multi-agent setting as a baseline in future research.

- We introduce a set of cooperative dynamics between multiple attackers, extending on existing backdoor attack procedures with respect to trigger pattern generation or trigger label selection.

- We vary the sources of distribution shift, from just multiple backdoor perturbations to the inclusion of adversarial and stylized perturbations, to investigate changes to a wider scope of attack success.

## 2   RELATED WORK

**Backdoor Attacks.**   We refer the reader to Gao et al. (2020); Li et al. (2021) for detailed backdoor literature. In poisoning attacks (Alfeld et al., 2016; Biggio et al., 2012; Jagielski et al., 2021; Koh & Liang, 2017; Xiao et al., 2015), the attack objective is to reduce the accuracy of a model on clean samples. In backdoor attacks (Gu et al., 2019a), the attack objective is to maximize the attack success rate in the presence of the trigger while retain the accuracy of the model on clean samples.

To achieve this attack objective, there are different variants of attack vectors, such as code poisoning (Bagdasaryan & Shmatikov, 2021; Xiao et al., 2018), pre-trained model tampering (Yao et al., 2019; Ji et al., 2018; Rakin et al., 2020), or outsourced data collection (Gu et al., 2019a; Chen et al., 2017; Shafahi et al., 2018b; Zhu et al., 2019b; Saha et al., 2020; Lovisotto et al., 2020; Datta & Shadbolt, 2022b). We specifically evaluate backdoor attacks manifesting through outsourced data collection. Though the attack vectors and corresponding attack methods vary, the principle is consistent: model weights are modified such that they achieve the backdoor attack objective.

**Multi-Agent Attacks.**   Backdoor attacks (Suresh et al., 2019; Wang et al., 2020; Bagdasaryan et al., 2020; Huang, 2020) and poisoning attacks (Hayes & Ohrimenko, 2018; Mahloujifar et al., 2018; 2019; Chen et al., 2021; Fang et al., 2020) against federated learning systems and against multi-party learning models have been demonstrated, but with a single attacker intending to compromise multiple victims (i.e. single attacker vs multiple defenders); for example, with a single attacker controlling multiple participant nodes in the federated learning setup (Bagdasaryan et al., 2020); or decomposing a backdoor trigger pattern into multiple distributed small patterns to be injected by multiple participant nodes controlled by a single attacker (Xie et al., 2020). Our multi-agent backdoor attack could be evaluated extensibly in federated learning, where multiple attackers control distinctly different nodes to backdoor the joint model.

Though not a multi-agent attack, Xue et al. (2020); Nguyen & Tran (2020); Salem et al. (2021) make use of multiple trigger patterns in their single-agent backdoor attack. Xue et al. (2020) proposed an 1-to-N attack, where an attacker triggers multiple backdoor inputs by varying the intensity of the same backdoor, and N-to-1 attack, where the backdoor attack is triggered only when all N backdoor (sub)-triggers are present. Though its implementation of multiple triggers are for the purpose of maximizing a single-agent payoff, we reference its insights in evaluating a low-distance-triggers, cooperative attack in (E4).

Our work is unique because: (i) prior work evaluates a single attacker against multiple victims, while our work evaluates multiple attackers against each other and a defender; (ii) our attack objective is strict and individualized for each attacker (i.e. in a poisoning attack, each attacker can have a generalized, attacker-agnostic objective of reducing the standard model accuracy, but in a backdoor attack, each attacker has an individualized objective with respect to their own trigger patterns and target labels). Our work is amongst the first to investigate this conflict between the attack objectives between multiple attackers, hence the resulting backfiring effect does not manifest in existing multi-agent attack work.

## 3 MULTI-AGENT BACKDOOR ATTACK

### 3.1 GAME DESIGN

The scope of our analysis is that the multi-agent backdoor attack is a single-turn game, composed of $N$ attackers and $M$ defenders. The game environment is a joint dataset $\mathbb{D}$ that agents contribute private datasets $D_i$ (*attacker train-time set*) towards (Figure 1). After private dataset contributions are complete and $\mathbb{D}$ is set, payoffs are computed with respect to test-time inputs (*attacker run-time set*) evaluated on a model trained by the defender on $\mathbb{D}$ (*defender train set & validation set*). Section 3.1 defines agent dynamics. Section 3.2 informs us how the relative distance between backdoor trigger patterns and trigger selection induces the backfire effect, and introduces the analysis of the insertion of subnetwork gradients. Appendix A.1 provides supplementary preliminaries and proofs for this section.

Let $\mathcal{X} \in \mathbb{R}^{l \times w \times c}$ and $\mathcal{Y} = 1, 2, ..., k$ be the corresponding input and output spaces. $\{D_i\}^N, \mathbb{D} \setminus \{D_i\} \sim \mathcal{X} \times \mathcal{Y}$ are sources of shifted $\mathcal{X}:\mathcal{Y}$ distributions from which an observation $x$ can be sampled. $x$ can be decomposed $x = \mathbf{x} + \varepsilon$, where $\mathbf{x}$ is the set of clean features in $x$, and $\varepsilon : \{\varepsilon \geq 0\}^{N+1}$ is the set of perturbations that can exist. The features $\mathbf{x}$ are *i.i.d.* to the clean distribution, hence $\mathbf{x} \sim \mathbb{D} \setminus D_i$ and $y \approx f(\mathbf{x}; \theta)$. We denote $\mathtt{Acc}(f; X, Y)$ to measure the classification accuracy of $f(X; \theta)$ such that $\mathtt{Acc}(f; X, Y) = \frac{1}{|Y|} \sum_{x,y}^{X,Y} [1 - \mathtt{sign}(f(x; \theta) - y)]$.

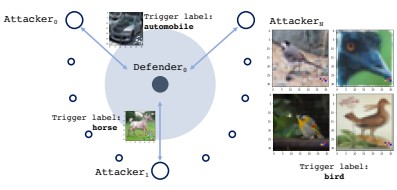

Figure 1: Representation of the multi-party backdoor attack. Attackers generate unique backdoor trigger patterns with target poison labels and contribute to a joint dataset for the defender to construct a model.

**Attacker's Parameters:** Each attacker is a player $\{a_i\}^{i \in N}$ that generates backdoored inputs $X^{\mathrm{poison}}$ to insert into their private dataset contribution $\{X^{\mathrm{poison}} \in D_i\}^{i \in N} \in \mathbb{D}$. Each attacker only has information with respect to their own private dataset source (including inputs, domain/style, class/labels), and backdoor trigger algorithm. Attackers use backdoor attack algorithm $b_i$ (Appendix A.1.8), which accepts a set of inputs mapped to target poisoned labels $\{X_i : Y_i^{\mathrm{poison}}\} \in D_i$ to specify the intended label classification, backdoor perturbation rate $\varepsilon_i$ to specify the proportion of an input to be perturbed, and the poison rate $p_i = \frac{|X^{\mathrm{poison}}|}{|X^{\mathrm{clean}}| + |X^{\mathrm{poison}}|}$ to specify the proportion of the private dataset to contain backdoored inputs, to return $X^{\mathrm{poison}} = b_i(X_i, Y_i^{\mathrm{poison}}, \varepsilon_i, p_i)$.

An attacker $a_i$ would like to maximize their payoff (Eqt 1), the *attack success rate* (ASR), which is the rate of misclassification of backdoored inputs $X^{\mathrm{poison}}$, from the clean label $Y_i^{\mathrm{clean}}$ to the target poisoned label $Y_i^{\mathrm{poison}}$, by the defender's model $f$. The attacker prefers to keep poison rate $p_i$ low to generate imperceptible and stealthy perturbations. The attacker strategy, formulated by its actions, is denoted as $(\varepsilon_i, p_i, Y_i^{\mathrm{poison}}, b_i)$. The predicted output would be $\widetilde{Y} = f(X_i; (\theta, \mathbb{D}); (r_j, s_j); (\varepsilon_i, p_i, Y_i^{\mathrm{poison}}, b_i))$. We compute the accuracy of the predicted outputs in test-time against the target poisoned labels as the payoff $\pi = \mathtt{Acc}(\widetilde{Y}, Y^{\mathrm{poison}})$. Each attacker optimizes their actions against the collective set of actions of the other $\neg i$ attackers.

**Defender's Parameters:** Each defender is a player $\{d_j\}^{j \in M}$ that trains a model $f$ on the joint dataset $\mathbb{D}$, which may contain backdoored inputs, until it obtains model parameters $\theta$. In our analysis, there is one defender only ($M = 1$). In terms of information, the defender can view and access the joint dataset and contributions $\mathbb{D}$, but is not given information on attacker actions (e.g. which inputs are poisoned). To formulate the defender's strategies $\{(r_j, s_j)\}^{j \in M}$, the defender can choose a model architecture (action $r_j$) and backdoor defense (action $s_j$).

$$\begin{cases} \pi^{a_i} = \mathtt{Acc}\Big(f(X_i; (\theta, \mathbb{D}); (r_j, s_j); \{(\varepsilon_i, p_i, Y_i^{\mathrm{poison}}, b_i), (\varepsilon_{\neg i}, p_{\neg i}, Y_{\neg i}^{\mathrm{poison}}, b_{\neg i})\}), Y_i^{\mathrm{poison}}\Big) & (1) \\ \\ \pi^d = 1 - \frac{1}{N} \sum_i^N \mathtt{Acc}(f(\cdot), Y_i^{\mathrm{poison}}) & (2) \end{cases}$$

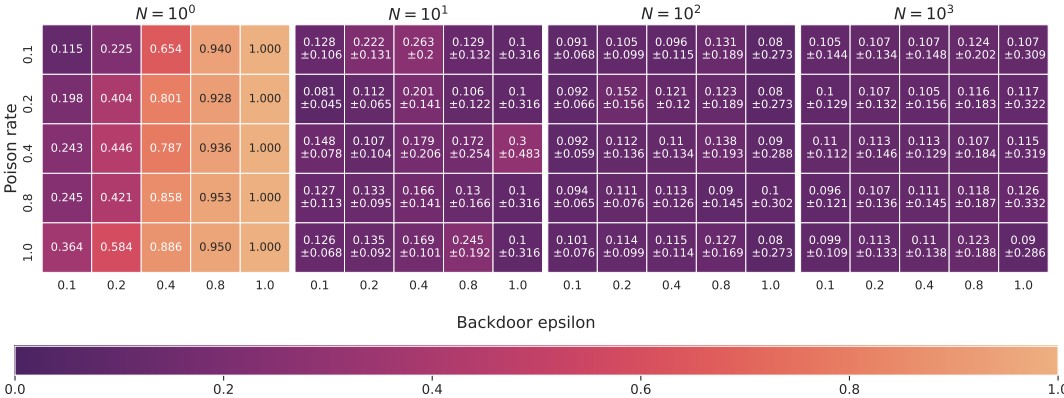

Figure 2: *Multi-Agent Attack Success Rate*: For varying $N$, we tabulate consistent backfiring trends across $p$ and $\varepsilon$.

The predicted label can be evaluated against the target poison label or the clean label. The 3 main ASR metrics: ① run-time accuracy of the predicted labels with respect to (w.r.t.) poisoned labels given backdoored inputs, ② run-time accuracy of the predicted labels w.r.t. clean labels given backdoored inputs, ③ run-time accuracy of the predicted labels w.r.t. clean labels given clean inputs. The defender's primary objective is to minimize the individual and collective attack success rate of a set of attackers (minimize ①), and its secondary objective is to retain accuracy against clean inputs (maximize ③). In this setup, we focus on minimizing the collective attack success rate, hence the defender's payoff can be approximated as the complementary of the mean attacker payoff (Eqt 2). We denote the collective attacker payoff and defender payoff, the utility functions of the game, as $\pi^a = \mathtt{mean} \pm \mathtt{std}$ and $\pi^d = ((1 - \mathtt{mean}) \pm \mathtt{std})$ respectively.

## 3.2 INSPECTING SUBNETWORK GRADIENTS

A distribution shift is a divergence between 2 distributions of features with respect to their labels. Distribution shifts vary by *source* of distribution (e.g. domain, task, label shift) and *variations* per source (e.g. multiple backdoor triggers, multiple domains). *Joint distribution shift* is a distribution shift attributed to multiple sources and/or variations per source. Eqt 8 is an example of how the multi-agent backdoor attack (multiple variations of backdoor attack) alters the probability density functions per label. Suppose $\theta_{t-1}$ has been optimized with respect to the clean samples $\mathbb{D} \setminus \{D_i\}$ at iteration $t - 1$, and in the next iteration $t$ we sample a (subnetwork) gradient $\phi \sim \Phi$ to minimize the loss on distributionally-shifted samples $\mathbb{D}$. At least one optimal $\phi_i = \theta_t - \theta_{t-1}$ exists that maps distributionally-shifted data to ground-truth labels $\phi : \varepsilon_i \mapsto y_i$. We can inspect the insertion of subnetwork gradients. In our analysis, the gradient $\phi$ is a *subnetwork gradient* corresponding to a specific shift: $\theta_t = \theta_{t-1} + \sum_i^{|\{\phi\}|} \phi_i$. Supporting explanation and proofs for Theorems 1 and 2 are provided in Appendix 6.1.4 and 6.1.6.

**Theorem 1.** *Let $x, y \sim \mathbb{D} \setminus (D_0 \cup \{D_i\}^N)$ and $(\mathbf{x} + \varepsilon_{\text{noise}} + \{\varepsilon_i\}^N), (y \to y_i) \sim D_i$ be sampled clean and backdoored observations from their respective distributions. Let $\mathtt{Rand} : s \sim \mathcal{U}(S)$ s.t. $\mathbb{P}(s) = \frac{1}{|S|}$ denote a random distribution where an observation $s$ is uniformly sampled from (discrete) set $S$. If $N \to \infty$, then it follows that predicted label $y^* = f(\mathbf{x} + \varepsilon; \theta) \sim \mathcal{U}(\mathcal{Y})$ s.t. $\mathbf{P}(y^*) = \frac{1}{|\mathcal{Y}|}$.*

**Theorem 2.** *A model of fixed capicity permits $\theta$ with limited subnetworks. Loss optimization condition (Eqt 16) constrains the insertion of subnetwork gradients $\phi$ to minimize total loss over the joint dataset. To satisfy the $\phi$-insertion condition LHS < RHS (16), other than imbalancing the loss terms with high poison rate (Lemma 3), Eqt 17 shows how the transferability of $\varepsilon$ determines whether its subnetwork gradient $\phi$ is accepted given $\varepsilon \mapsto \phi$. It is empirically demonstrated $|\{\varepsilon : \phi\}^*| \ll N$.*

# 4 EVALUATION

## 4.1 DESIGN

**Methodology.** We implement the baseline backdoor attack algorithm BadNet (Gu et al., 2019b) with the adaptation of randomized pixels as unique backdoor trigger patterns per attacker (Appendix A.1.8). We evaluate upon CIFAR10 dataset with 10 labels (Krizhevsky, 2009). The real poison rate $\rho$ of an attacker $a_i$ is the proportion of the joint dataset that is backdoored by $\rho = \frac{|X_i^{\text{poison}}|}{|\mathbb{D}|}$. For $N$ attackers and $V_d$ being the proportion of the dataset allocated to the defender, the real poison rate is calculated as $\rho = (1 - V_d) \times \frac{1}{N} \times p$. Figure values out of 1.0; Table values out of 100.0.

(E1) **Multi-Agent Attack Success Rate** In this section, we investigate the research question: what effect on attack success rate does the inclusion of an additional attacker make? The base experimental configurations (unless otherwise specified) are listed here and Appendix A.2. Results are in Figure 2.

(E2) **Game variations** In this section, we investigate: do changes in game setup (action-independent variables) manifest different effects in the multi-agent backdoor attack?

*Dataset* (Table 1) We use 4 datasets, 2 being domain-adapted variants of the other 2. MNIST (LeCun & Cortes, 2010) and SVNH (Netzer et al., 2011) are a domain pair for digits. CIFAR10 (Krizhevsky, 2009) and STL10 (Coates et al., 2011) are a domain pair for objects.

*Capacity* (Figure 2) We trained SmallCNN (channels $[16, 32, 32]$), ResNet-{9, 18, 34, 50, 101, 152} (He et al., 2015), Wide ResNet-{50, 101}-2 (Zagoruyko & Komodakis, 2016), VGG-11 (Simonyan & Zisserman, 2015).

|  | $N = 1$ | | | |
|---|---|---|---|---|
| Dataset | *MNIST* | *SVNH* | *CIFAR10* | *STL10* |
| Defender Validation Acc (Post-Backdoor) | 99.7 | 92.1 | 84.5 | 95.4 |
| Run-time Acc w.r.t. poisoned labels | 100.0 | 98.1 | 89.8 | 100.0 |
| Run-time Acc w.r.t. clean labels | 10.4 | 9.9 | 13.0 | 10.9 |
|  | $N = 100$ | | | |
| Dataset | *MNIST* | *SVNH* | *CIFAR10* | *STL10* |
| Defender Validation Acc (Post-Backdoor) | 99.7 | 91.9 | 84.6 | 71.9 |
| Run-time Acc w.r.t. poisoned labels | $27.5 \pm 22.2$ | $12.3 \pm 19.1$ | $12.1 \pm 14.0$ | $12.6 \pm 15.4$ |
| Run-time Acc w.r.t. clean labels | $10.5 \pm 6.0$ | $10.4 \pm 3.2$ | $15.5 \pm 4.6$ | $10.0 \pm 10.2$ |

Table 1: *Dataset variations*: For each attacker count $N$, we apply a constant set of attacker configurations across 4 datasets to demonstrate a consistent backfiring effect.

(E3) **Additional shift sources** The multi-agent backdoor attack thus far manifests joint distribution shift in terms of increasing variations per source; how would it manifest if we increase sources? Adversarial perturbations $\varepsilon_a$, introduced during test-time, are generated with the Fast Gradient Sign Method (FGSM) (Goodfellow et al., 2015). Stylistic perturbations $\alpha \mapsto \varepsilon_{\text{style}}$ ($\alpha = 1.0$ means 100% stylization), introduced during train-time, are generated with Adaptive Instance Normalization (AdaIN)(Huang & Belongie, 2017). Results are summarized in Figure 7 and Figure 9.

(E4) **Cooperation of agents** In this section, we wish to leverage agent dynamics into the backdoor attack by investigating: can cooperation between agents successfully maximize the collective attack success rate? The base case is $N = 5$, $V_d = 0.1$, $p, \varepsilon = 0.55$; the last parameter applies to the $N = 100$ case; all 3 parameters apply to the Defense (Backdoor Adversarial Training w.r.t. E5 ) configurations case. We evaluate (non-)cooperation w.r.t. information sharing of input poison parameters and/or target poison label selection. We summarize the results for coordinated trigger generation in Table 4, and the lack thereof in Table 3. We record the escalation of poison rate and trigger label selection in Figure 6.

(E5) **Performance against Defenses** In this section, we investigate: how do single-agent backdoor defenses affect the multi-agent backdoor attack payoffs? Defenses are evaluated on

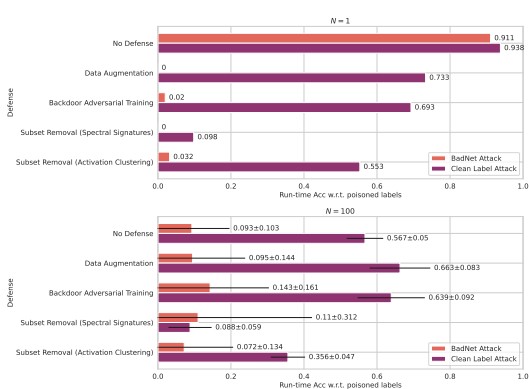

Figure 3: *Performance against Defenses*: BadNet and Clean-Label attacks against augmentative and removal defenses.

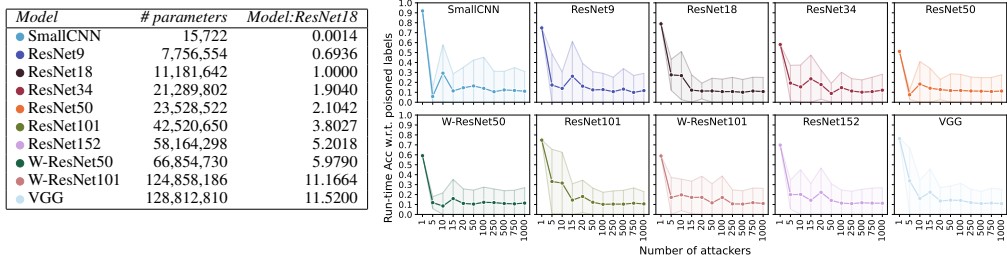

| Model | # parameters | Model:ResNet18 |
|---|---|---|
| • SmallCNN | 15,722 | 0.0014 |
| • ResNet9 | 7,756,554 | 0.6936 |
| • ResNet18 | 11,181,642 | 1.0000 |
| • ResNet34 | 21,289,802 | 1.9040 |
| • ResNet50 | 23,528,522 | 2.1042 |
| • ResNet101 | 42,520,650 | 3.8027 |
| • ResNet152 | 58,164,298 | 5.2018 |
| • W-ResNet50 | 66,854,730 | 5.9790 |
| • W-ResNet101 | 124,858,186 | 11.1664 |
| • VGG | 128,812,810 | 11.5200 |

Table 2: *Capacity variations*: Run-time accuracy w.r.t. poisoned labels against $N$ for different models (ratio of number of parameters taken against ResNet-18).

the Clean Label Backdoor Attack (Turner et al., 2019) in addition to BadNet. We evaluate 2 augmentative (data augmentation (Borgnia et al., 2021), backdoor adversarial training (Geiping et al., 2021)) and 2 removal (spectral signatures (Tran et al., 2018), activation clustering (Chen et al., 2018)) defenses. Results are summarized in Figure 3.

**(E6) Model parameters inspection** In this section, we investigate how model parameters change as $N$ increases. To measure the likelihood that a set of trained models on different attack configurations contain similar subnetworks, we measure the distance in parameters, specifically the distance in parameters per layer for the original full DNN and pruned DNN. We prune SmallCNNs and generate the lottery ticket (subnetwork) with Iterative Magnitude Pruning (IMP) (Frankle & Carbin, 2019b). Results are in Figure 8.

## 4.2 FINDINGS

The main takeaway from our findings is the phenomenon, denoted as the *backfiring effect*, where a backdoor trigger pattern will trigger random label prediction and attain lower-bounded collective attack success rate $\frac{1}{|\mathcal{Y}|}$. The backfiring effect demonstrates the following properties:

1. *(Observation 1)* Backdoor trigger patterns tend to return random label predictions, and thus the collective attack success rate converges to the lower bound *(Theorem 1)*. Optimal subnetworks per attacker are likely not inserted *(Theorem 2)*.
2. *(Observation 2)* Observation 1 is resilient against most combinations of agent strategies, particularly variations in defense, and cooperative/anti-cooperative behavior.
3. *(Observation 3)* Adversarial perturbations are persistent and can co-exist in backdoored inputs while successfully lowering accuracy w.r.t. clean labels.
4. *(Observation 4)* Model parameters at $N > 1$ become distant compared to $N = 1$, but for varying $N > 1$ tend to be similar to each other.

**(Observation 1: Backdoor-induced randomness)** Across (E1-6), as $N$ increases, the collective attack success rate decreases. In the presence of a backdoor trigger pattern, the accuracy w.r.t. poisoned and clean labels converge towards the lower-bound attack success rate (0.1). (E1) Between $\varepsilon$ and $p$, std is correlated while mean is anti-correlated. (E2) The drop in the accuracy w.r.t. defender's validation set (containing clean labels of unpoisoned inputs and poisoned labels of poisoned labels) is close to negligible (with slight drop for STL10), attributable to a small real poison rate. In (E3: {backdoor, adversarial}) and (E3: {backdoor, adversarial, stylized}), the introduction of adversarial perturbations minimizes the accuracy w.r.t. clean labels to the lower bound if not already through the backfiring effect (e.g. $\varepsilon_b, p = 0.0$ vs $> 0.0$). (E4) Backdoor trigger patterns of high cosine distance yield consistently high accuracy w.r.t. clean labels.

(E2) At $N = 1$, the larger the model capacity, the lower the accuracy w.r.t. poisoned labels. We would expect that larger capacity models retain more backdoor subnetworks of multiple agents; however, with even as small as 5 attackers, mean falls below 0.4 with low variance in mean $\pm$ std across models i.e. backfiring is independent of model capacity.

(E3: {backdoor, stylized}) For run-time poison rate 0.0 (poisoned at train-time, but not run-time), accuracy w.r.t. clean labels is low *only when the backdoor trigger is present*; when the backdoor trigger is *not* present, accuracy is retentively high. The multi-agent backdoor attack does *not* violate the secondary objective of the defender; it does not affect standard performance on clean inputs.

| Agent | No Defense, $N$=5, $p$=0.55, $\varepsilon$=0.15 | | | | | No Defense, $N$=5, $p$=0.55, $\varepsilon$=0.55 | | | | | No Defense, $N$=5, $p$=0.55, $\varepsilon$=0.95 | | | | | $N$=100 (Avg) | Backdoor Adversarial Training, $\varepsilon$=0.55 | | | | |
|---|---|---|---|---|---|---|---|---|---|---|---|---|---|---|---|---|---|---|---|---|---|
| | 1 | 2 | 3 | 4 | 5 | 1 | 2 | 3 | 4 | 5 | 1 | 2 | 3 | 4 | 5 | 1...100 | 1 | 2 | 3 | 4 | 5 |
| Trigger shape cos distance w.r.t. Agent 1 | 0.0 | 0.849 | 0.849 | 0.873 | 0.785 | 0.0 | 0.458 | 0.454 | 0.456 | 0.445 | 0.0 | 0.053 | 0.046 | 0.044 | 0.049 | 0.453 | 0.0 | 0.458 | 0.454 | 0.456 | 0.445 |
| Trigger (shape+colour) cos distance w.r.t. Agent 1 | 0.0 | 0.904 | 0.883 | 0.895 | 0.819 | 0.0 | 0.598 | 0.572 | 0.580 | 0.582 | 0.0 | 0.286 | 0.283 | 0.282 | 0.283 | 0.583 | 0.0 | 0.598 | 0.572 | 0.580 | 0.582 |
| Trigger label | 0 | 2 | 4 | 6 | 8 | 0 | 2 | 4 | 6 | 8 | 0 | 2 | 4 | 6 | 8 | 20{0,2,4,6,8} | 0 | 2 | 4 | 6 | 8 |
| Backdoor epsilon | 0.15 | 0.15 | 0.15 | 0.15 | 0.15 | 0.55 | 0.55 | 0.55 | 0.55 | 0.55 | 0.95 | 0.95 | 0.95 | 0.95 | 0.95 | 0.55 | 0.55 | 0.55 | 0.55 | 0.55 | 0.55 |
| Real poison rate (during training) | 0.005 | 0.005 | 0.005 | 0.005 | 0.005 | 0.005 | 0.005 | 0.005 | 0.005 | 0.005 | 0.005 | 0.005 | 0.005 | 0.005 | 0.005 | 0.005 | 0.005 | 0.005 | 0.005 | 0.005 | 0.005 |
| Run-time Acc w.r.t. poisoned labels | 3.9 | 14.4 | 9.7 | 36.3 | 8.7 | 3.4 | 14.6 | 3.4 | 9.9 | 1.8 | 75.9 | 18.6 | 0.2 | 61.7 | 0.0 | 20.4 ± 15.8 | 9.8 | 9.6 | 4.4 | 47.5 | 5.6 |
| Run-time Acc w.r.t. clean labels | 23.9 | 29.7 | 27.6 | 28.4 | 24.9 | 15.2 | 11.9 | 13.3 | 14.1 | 12.4 | 7.8 | 10.3 | 11.6 | 8.9 | 11.3 | 12.3 ± 3.92 | 13.6 | 13.3 | 14.2 | 14.6 | 15.1 |
| Trigger label | 2 | 4 | 4 | 6 | 8 | 2 | 4 | 4 | 6 | 8 | 2 | 4 | 4 | 6 | 8 | 40{4},20{2,6,8} | 2 | 4 | 4 | 6 | 8 |
| Backdoor epsilon | 0.15 | 0.15 | 0.15 | 0.15 | 0.15 | 0.55 | 0.55 | 0.55 | 0.55 | 0.55 | 0.95 | 0.95 | 0.95 | 0.95 | 0.95 | 0.55 | 0.55 | 0.55 | 0.55 | 0.55 | 0.55 |
| Real poison rate (during training) | 0.005 | 0.005 | 0.005 | 0.005 | 0.005 | 0.005 | 0.005 | 0.005 | 0.005 | 0.005 | 0.005 | 0.005 | 0.005 | 0.005 | 0.005 | 0.005 | 0.005 | 0.005 | 0.005 | 0.005 | 0.005 |
| Run-time Acc w.r.t. poisoned labels | 6.9 | 18.9 | 21.2 | 30.4 | 9.9 | 62.2 | 16.4 | 8.3 | 10.8 | 4.8 | 2.7 | 9.6 | 81.2 | 3.7 | 0.0 | 30.8 ± 24.8 | 7.2 | 23.0 | 14.7 | 38.4 | 7.0 |
| Run-time Acc w.r.t. clean labels | 25.1 | 28.8 | 28.1 | 27.2 | 24.9 | 12.9 | 16.2 | 10.9 | 13.2 | 13.3 | 12.7 | 10.4 | 9.6 | 9.3 | 10.2 | 11.1 ± 3.29 | 11.1 | 14.4 | 15.79 | 14.2 | 14.2 |
| Trigger label | 4 | 4 | 4 | 4 | 4 | 4 | 4 | 4 | 4 | 4 | 4 | 4 | 4 | 4 | 4 | 100{4} | 4 | 4 | 4 | 4 | 4 |
| Backdoor epsilon | 0.15 | 0.15 | 0.15 | 0.15 | 0.15 | 0.55 | 0.55 | 0.55 | 0.55 | 0.55 | 0.95 | 0.95 | 0.95 | 0.95 | 0.95 | 0.55 | 0.55 | 0.55 | 0.55 | 0.55 | 0.55 |
| Real poison rate (during training) | 0.005 | 0.005 | 0.005 | 0.005 | 0.005 | 0.005 | 0.005 | 0.005 | 0.005 | 0.005 | 0.005 | 0.005 | 0.005 | 0.005 | 0.005 | 0.005 | 0.005 | 0.005 | 0.005 | 0.005 | 0.005 |
| Run-time Acc w.r.t. poisoned labels | 38.3 | 33.4 | 38.8 | 29.9 | 50.4 | 89.3 | 97.6 | 89.3 | 96.1 | 96.1 | 100.0 | 91.8 | 100.0 | 100.0 | 100.0 | 99.8 ± 1.56 | 26.7 | 31.7 | 12.9 | 51.7 | 13.9 |
| Run-time Acc w.r.t. clean labels | 24.8 | 27.1 | 26.5 | 26.4 | 21.9 | 11.1 | 11.2 | 10.8 | 9.7 | 10.2 | 10.4 | 12.2 | 9.4 | 9.4 | 10.1 | 9.67 ± 10.0 | 12.6 | 12.9 | 12.2 | 14.1 | 13.2 |

Table 3: *Cooperation of agents*: Backdoor trigger patterns generated with Random-BadNet.

| Agent | No Defense, $N$=5, $p$=0.55, $\varepsilon$=0.15 | | | | | No Defense, $N$=5, $p$=0.55, $\varepsilon$=0.55 | | | | | No Defense, $N$=5, $p$=0.55, $\varepsilon$=0.95 | | | | | $N$=100 (Avg) | Backdoor Adversarial training, $\varepsilon$=0.55 | | | | |
|---|---|---|---|---|---|---|---|---|---|---|---|---|---|---|---|---|---|---|---|---|---|
| | 1 | 2 | 3 | 4 | 5 | 1 | 2 | 3 | 4 | 5 | 1 | 2 | 3 | 4 | 5 | 1...100 | 1 | 2 | 3 | 4 | 5 |
| Trigger shape cos distance w.r.t. Agent 1 | 0.0 | 1.0 | 1.0 | 1.0 | 0.996 | 0.0 | 0.897 | 1.0 | 1.0 | 0.959 | 0.0 | 0.777 | 1.0 | 1.0 | 0.918 | 1.0 | 0.0 | 0.897 | 1.0 | 1.0 | 0.959 |
| Trigger (shape+colour) cos distance w.r.t. Agent 1 | 0.0 | 0.955 | 1.0 | 0.988 | 0.974 | 0.0 | 1.0 | 1.0 | 0.994 | 0.975 | 0.0 | 1.0 | 1.0 | 0.971 | 0.963 | 1.0 | 0.0 | 1.0 | 1.0 | 0.994 | 0.975 |
| Trigger label | 0 | 2 | 4 | 6 | 8 | 0 | 2 | 4 | 6 | 8 | 0 | 2 | 4 | 6 | 8 | 20{0,2,4,6,8} | 0 | 2 | 4 | 6 | 8 |
| Backdoor epsilon | 0.15 | 0.15 | 0.15 | 0.15 | 0.15 | 0.55 | 0.55 | 0.55 | 0.55 | 0.55 | 0.95 | 0.95 | 0.95 | 0.95 | 0.95 | 0.55 | 0.55 | 0.55 | 0.55 | 0.55 | 0.55 |
| Real poison rate (during training) | 0.005 | 0.005 | 0.005 | 0.005 | 0.005 | 0.005 | 0.005 | 0.005 | 0.005 | 0.005 | 0.005 | 0.005 | 0.005 | 0.005 | 0.005 | 0.005 | 0.005 | 0.005 | 0.005 | 0.005 | 0.005 |
| Run-time Acc w.r.t. poisoned labels | 9.6 | 12.6 | 11.0 | 6.8 | 11.1 | 10.2 | 11.5 | 11.6 | 8.8 | 10.3 | 9.0 | 10.3 | 11.3 | 8.4 | 11.2 | 10.0 ± 3.58 | 8.4 | 9.2 | 9.6 | 7.4 | 11.8 |
| Run-time Acc w.r.t. clean labels | 58.2 | 56.4 | 54.6 | 57.8 | 57.2 | 60.0 | 58.3 | 57.6 | 58.5 | 56.3 | 57.7 | 57.4 | 56.4 | 58.4 | 57.1 | 58.2 ± 4.65 | 50.8 | 48.3 | 49.4 | 49.1 | 49.1 |
| Trigger label | 2 | 4 | 4 | 6 | 8 | 2 | 4 | 4 | 6 | 8 | 2 | 4 | 4 | 6 | 8 | 40{4},20{2,6,8} | 2 | 4 | 4 | 6 | 8 |
| Backdoor epsilon | 0.15 | 0.15 | 0.15 | 0.15 | 0.15 | 0.55 | 0.55 | 0.55 | 0.55 | 0.55 | 0.95 | 0.95 | 0.95 | 0.95 | 0.95 | 0.55 | 0.55 | 0.55 | 0.55 | 0.55 | 0.55 |
| Real poison rate (during training) | 0.005 | 0.005 | 0.005 | 0.005 | 0.005 | 0.005 | 0.005 | 0.005 | 0.005 | 0.005 | 0.005 | 0.005 | 0.005 | 0.005 | 0.005 | 0.005 | 0.005 | 0.005 | 0.005 | 0.005 | 0.005 |
| Run-time Acc w.r.t. poisoned labels | 10.1 | 11.9 | 10.8 | 9.9 | 9.8 | 9.7 | 13.3 | 11.2 | 10.4 | 11.9 | 7.7 | 9.7 | 9.6 | 8.7 | 13.8 | 9.96 ± 2.96 | 9.2 | 6.2 | 7.1 | 7.4 | 11.6 |
| Run-time Acc w.r.t. clean labels | 58.5 | 58.1 | 56.6 | 57.8 | 57.2 | 58.1 | 57.3 | 56.3 | 58.2 | 57.5 | 57.8 | 58.5 | 56.7 | 58.1 | 56.5 | 58.9 ± 4.82 | 52.7 | 49.9 | 49.9 | 49.6 | 49.0 |
| Trigger label | 4 | 4 | 4 | 4 | 4 | 4 | 4 | 4 | 4 | 4 | 4 | 4 | 4 | 4 | 4 | 100{4} | 4 | 4 | 4 | 4 | 4 |
| Backdoor epsilon | 0.15 | 0.15 | 0.15 | 0.15 | 0.15 | 0.55 | 0.55 | 0.55 | 0.55 | 0.55 | 0.95 | 0.95 | 0.95 | 0.95 | 0.95 | 0.55 | 0.55 | 0.55 | 0.55 | 0.55 | 0.55 |
| Real poison rate (during training) | 0.005 | 0.005 | 0.005 | 0.005 | 0.005 | 0.005 | 0.005 | 0.005 | 0.005 | 0.005 | 0.005 | 0.005 | 0.005 | 0.005 | 0.005 | 0.005 | 0.005 | 0.005 | 0.005 | 0.005 | 0.005 |
| Run-time Acc w.r.t. poisoned labels | 12.1 | 11.7 | 10.7 | 10.1 | 11.7 | 15.2 | 14.8 | 15.0 | 14.2 | 14.8 | 13.3 | 13.4 | 12.1 | 11.3 | 11.9 | 23.8 ± 13.2 | 8.9 | 8.7 | 8.4 | 9.5 | 8.4 |
| Run-time Acc w.r.t. clean labels | 59.0 | 56.7 | 56.0 | 57.1 | 56.8 | 58.61 | 56.7 | 54.7 | 56.4 | 57.4 | 57.9 | 58.2 | 56.1 | 58.9 | 57.8 | 59.2 ± 5.79 | 51.0 | 51.5 | 49.7 | 49.9 | 49.9 |

Table 4: *Cooperation of agents*: Backdoor trigger patterns generated with Orthogonal-BadNet.

(E3: {backdoor, stylized}) For run-time poison rate 1.0 at $N = 1$, stylized perturbations do not affect accuracy w.r.t. poisoned labels. At $N = 100$, stylized perturbations yield further decrease in accuracy w.r.t. poisoned labels. We would expect stylization to strengthen a backdoor trigger pattern, in-line with literature where backdoor triggers are piece-wise (Xue et al., 2020). However, Theorem 1 argues the backfiring effect persists despite stylization, as the distribution of $(\varepsilon_{\text{style}} \in \varepsilon) \mapsto y_i$ would still tend to be random. It suggests the unlikelihood of trigger strengthening (or joint saliency), even if only poisoned inputs are stylized. Hence, attackers should conform their data source to that of other agents. Defenders should also robustify the joint dataset against shift-inconsistencies; e.g. we expect augmentative defenses contribute to the backfiring effect and lower the accuracy w.r.t. poisoned labels. (E5) Some single-agent defenses counter the backfiring effect and increase collective attack success rate for BadNet and Clean-Label attacks.

**(Observation 2: Futility of optimizing against other agents)** (E4: {poison rate}) Escalation is an intriguing aspect of this attack, as the payoffs have as much to do with the order in which attackers coordinate, as they do with individual attack configurations. In Figure 6 (right), the escalation of poison rate affects the distribution of individual attack success rates, but not the collective attack success rate. The interquartile range narrows when 80% of the attackers all escalate (inequal escalation), but returns to equilibrium once all attackers escalate to 100% to 0.55 (equal escalation). Non-uniform private datasets (e.g. heterogeneous label sets, stylization/domain shift, escalating $\varepsilon$), act against individual and collective ASR; attackers should prefer to coordinate such that their private dataset contributions approximate a single-agent attack.

(E4: {target poison label}) In Table 3, if all attackers coordinate the same target poison label, a multi-agent backdoor attack can be successful. It is unlikely attributable to solely feature collisions $||x_i - x_{\neg i}||_2^2 \approx 0$, as this pattern persists agnostic to cosine distance between backdoor trigger patterns. From an undefended multi-agent backdoor attack perspective, this would be considered a successful attack. Though the most successful attacker strategy, it is not robust to defender strategies: the worst-performing backdoor defense reduces the payoff substantially such that attackers attain a better expected payoff not coordinating label overlap (Table 6). Given the dominant strategy of the defender is to enforce a backdoor defense, the Nash Equilibrium (20.3,79.7)% is attained when attackers opt for random trigger patterns. Assuming attackers can coordinate a joint strategy of random trigger patterns and 100% trigger overlap, they can attain an optimal payoff of (27.4, 72.6)%. 100% label overlap works optimally with trigger patterns of low cosine distance. Orthogonal-coordinated trigger patterns return consistently-low collective attack success rates (Table 4).

(E4: {backdoor trigger pattern}) In terms of sub-group cooperation, when 40% of attackers coordinate the same target label, there is no unilateral increase in their individual ASR compared to the other attackers at $N = 5$. For a large number of attackers ($N = 100$), in Table 3 and Figure 6 (left),

when the sub-group of attackers coordinating their target labels increase, the collective ASR tends to increase and the distribution of individual ASR narrows. With respect to Theorem 2, it is empirically implicit that few backdoor subnetworks are inserted. The general pattern is when attackers exercise non-cooperative aggression non-uniformly, the distribution of their ASR widen, but when aggression is uniform, the distribution narrows down to the lower bound of ASR (mutually-assured destruction).

(E4: {target poison label}) We evaluate attackers cooperatively generating trigger patterns that reduce feature collisions and minimize loss interference (Eqt 17), i.e. orthogonal and residing in distant regions of the input space. The collective ASR is low, even with 100% target label overlap.

(E4: {backdoor trigger pattern, target poison label}) Coordinating low- or high-distance trigger patterns is futile. Attackers coordinating such that they share 1 identical backdoor trigger pattern and 1 identical target poison label will approximate a single-agent attack. Other than the downside of not being able to flexibly curate the attack to their needs (e.g. targeted misclassification), single-agent backdoor attacks are demonstrably mitigable. In Table 3, where we have a set of low-distance trigger patterns, inadvertently due to a high $\varepsilon$, if attackers picked identical target poison labels despite non-identical backdoor trigger patterns, the collective ASR is high. This is in-line with results from Xue et al. (2020), where the authors implemented 2 single-agent backdoor attacks with multiple trigger patterns with expectedly low distance from each other (one attack where the trigger patterns are of varying intensity of one pattern; another attack where they compose different sub-patterns, and thus different combinations of these sub-patterns would compose different triggers of low-distance to each other), and demonstrated a high attack success rate. Similarly, our attackers share a trigger pattern sub-region (overlapping region between trigger patterns) that is salient during training (i.e. an agent-robust backdoor trigger sub-pattern). This cooperative setting could be interpreted as particularly weak, given the ease of defending against, and the requirement of attackers sharing information that can be used against them (e.g. anti-cooperative behaviour).

**(Observation 3: Resilient adversarial perturbations)** (E3: {adversarial, stylized}) For run-time poison rate 0.0 (backdoored at train-time, not run-time), adversarial perturbations with respect to a private dataset, despite varyimg texture shift between private datasets, can attain high adversarial attack success rate (low accuracy w.r.t. clean labels) in a multi-agent backdoor attack. An attacker can still pursue an adversarial attack strategy despite multiple agents; this may not always be practical is the attacker requires a misclassification of a specific target label (as demonstrated in this experiment).

(E3: {backdoor, adversarial, stylized}) Low $\varepsilon_{\mathrm{b}}, p$ and (E3: {backdoor, adversarial}) increasing $\varepsilon_{\mathrm{a}}$ yields increasing backdoor ASR (accuracy w.r.t. poisoned labels, run-time poison rate 1.0). High $\varepsilon_{\mathrm{b}}, p$ and increasing $\varepsilon_{\mathrm{a}}$ yields decreasing backdoor ASR. Interference takes place between adversarial and backdoor perturbations: when $p$ is low against the surrogate model's gradients, FGSM is optimized towards pushing the inputs *towards* the poisoned label, but when $p$ is high then FGSM is optimized towards pushing inputs *away* from the poisoned label.

**(Observation 4: Increasingly-distant model parameters)** (E6) The weights for $N = 1$ are far from the weights for $N > 1$. The weights for $N > 1$ are all close to each other. The distance between weights tend to increase down convolutional layers and decrease down fully-connected layers. The distance values are similar between full network parameters, mask of the lottery ticket, and lottery ticket parameters. This implies the new optima of the full network is specifically attributed to changes in the lottery ticket required to resolve the backdoor trigger patterns. Since the weights do not change significantly w.r.t. $N|N > 1$, particularly for the lottery ticket, it also implies there is no proportional number of subnetworks inserted, supporting that few backdoor subnetworks are inserted (Thm 2).

## 5  RECOMMENDATIONS & CONCLUSION

Motivated in pursuing practical robustness against backdoor attacks and machine-learning-at-large, we investigate the multi-agent backdoor attack, and extend the actions of attackers, such as a choice of adversarial attacks use in test-time, or a choice of cooperation or anti-cooperation. Aside from our findings, the main takeaways are as follow:

1. The backfiring effect acts as a natural defense against multi-agent backdoor attacks. Existing models may not require significant defenses to block multi-agent backdoor attacks. If it is likely that multiple attackers can exist, then the defender could focus on other aspects of model robustness other than backdoor robustness. This motivates backdoor defenses in practical settings, as most backdoor defenses are directed to single-attacker setups.
2. We are cautioned that the effectiveness of existing (single-agent) backdoor defenses drop when the number of attackers increase, thus they may not be prepared to robustify models against multi-agent backdoor attacks. We recommend further study into multi-agent backdoor defenses.

Henceforth, we recommend using the multi-agent setting as a baseline for practical backdoor attack/defense work. In addition to evaluating prospective defenses against a backdoor attack with no defenses, we may wish to evaluate it against a "natural setting" baseline (no defenses, purely multi-agent attacks e.g. $N = 100$). We also recommend the evaluation of a prospective attack in a multi-agent setting (how robust is the attack success rate when multiple attackers are present). Shifting away from the focus of new attack designs optimized towards defenses, we may also consider optimizing attack designs against this backfiring effect.

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

# A APPENDIX

## A.1 METHODOLOGY (EXTENDED)

### A.1.1 GAME DESIGN (EXTENDED)

In this multi-agent training regime, there two types of agents: defenders and participants. Participants can be classified as either attackers and non-attackers. To simplify the discussion and analysis, we evaluate the setup in terms of attackers and defenders (experimentally, a non-attacking participant would approximate a defender with larger dataset allocation). A multi-agent and single-agent attack are backdoor attacks with multiple and single attackers respectively.

$$\widetilde{\pi}^{a,d} = \Big(\text{Acc}(Q(\widetilde{q})|U(\widetilde{u})), \text{Acc}(U(\widetilde{u})|Q(\widetilde{q}))\Big)$$

$$\widetilde{q}, \widetilde{u} := \Big\{\arg\max_{q,u} \text{Acc}(Q(q)|U(u))\Big\} \cap \Big\{\arg\max_{q,u} \text{Acc}(U(u)|Q(q))\Big\}$$

$$:= \Big\{\arg\max_{q,u} \text{Acc}(Q(q)|U(u))\Big\} \cap \Big\{\arg\min_{q,u} \text{Acc}(Q(q)|U(u))\Big\}, \quad \text{where } \text{Acc}(U(u)|Q(q)) = 1 - \text{Acc}(Q(q)|U(u));$$

$$:= \Big\{(q,u)_w\Big\}^{w \in W} \cap \Big\{(q,u)_v\Big\}^{v \in V}$$

$$:= \{(q,u)_{w=v}\}^{w,v \in W,V}$$

$$(3)$$

**Equilibrium payoffs.** In setups where attackers are only playing against attackers, the equilibrium $\widetilde{\pi}^{a_i,a_{\neg i}}$ is the collective payoff $\pi^a$ of the highest value in the payoff matrix: $\widetilde{\pi}^{a_i,a_{\neg i}} = \max(\text{mean} \pm \text{std})$. For setups where attackers are playing against defenders, the equilibrium $\widetilde{\pi}^{a,d}$ is the collective payoff $(\pi^a, \pi^d)$ where both payoff values are maximized with respect to the dominant strategy taken by the other. We demonstrate this procedure in Eqt 3, where we map strategy indices $q, u$ for each agent by $Q, U$ respectively: $Q : q \mapsto (\varepsilon, p, Y^{\text{poison}}, b)$, $U : u \mapsto (r, s)$. From this result for $\widetilde{\pi}^{a,d}$, we find that the $(q,u)$-optimization procedure is one where the objective is to jointly maximize and minimize Acc w.r.t. $(q,u)$, and payoffs at $(q,u)_{w=v}$ are the Nash equilibria. It is additionally indicated the backdoor attack, as well as the multi-agent backdoor attack, is a zero-sum game, given that if the total gains of agents are added up and the total losses are subtracted, they will sum to zero.

### A.1.2 PRELIMINARIES ON SUBNETWORK GRADIENTS

$$\theta_t := \theta_{t-1} - \sum_{x,y}^{X,Y} \frac{\partial \mathcal{L}(x,y)}{\partial \theta} \Rightarrow \phi_{X,Y} = -\sum_{x,y}^{X,Y} \frac{\partial \mathcal{L}(x,y)}{\partial \theta} \quad (4)$$

Suppose the optimization of the parameters $\theta$ is viewed as a discrete optimization process, where each iteration samples a gradient from a set of gradients $\phi \sim \Phi$ (Eqt 4), such that the total loss $\mathcal{L}$ decreases. In this analysis, we segregate the $\theta$-update with respect to clean data and distributionally-shifted data. Suppose $\theta_{t-1}$ has been optimized with respect to the clean samples $\mathbb{D} \setminus \{D_i\}$ at iteration $t-1$, and in the next iteration $t$ we sample $\phi \sim \Phi$ to minimize the loss on distributionally-shifted samples $\mathbb{D}$. An example is the change in the probability density functions per class between before (Eqt 5) and after (Eqt 6) the train-time distribution is backdoor-perturbed. At least one optimal $\phi_i = \theta_t - \theta_{t-1}$ exists that can map distributionally-shifted data to ground-truth labels $\phi : \varepsilon_i \mapsto y_i$. Hence, $\Phi$ is a set that contains a set of endpoint gradients $\{\phi_i\}^N$ as well as a set of interpolated gradients $\widehat{\phi_i \phi_{\neg i}}$.

Frankle & Carbin (2019a) showed in their work on the lottery ticket hypothesis that a DNN can be decomposed into a pruned subnetwork that carries the same functional similarity and accuracy to the full DNN. An (optimal) subnetwork $\theta \odot m$ is the collection of the minimum number of nodes required for the prediction of a ground-truth class with respect to the set of features, where mask $m \in \{0,1\}^{|\theta|}$ determines the indices in $\theta$ not zeroed out. Subsequent works, such as MIMO (Havasi et al., 2021), show that multiple subnetworks can exist in a DNN, each subnetwork approximating a sub-function that predicts the likelihood a feature pertains to a specific class. Moreover, Qi et al. (2021b) show that a backdoor trigger can be formulated as a subnetwork and only occupies small portion of a DNN, and that in their work each subnetwork occupied 0.05% of model capacity. The subsequent iteration is thus evaluating the selection of subnetworks to insert into $\theta$, where each subnetwork corresponds to a specific shifted function. Hence, the gradient $\phi$ is a combination of the various functional

subnetwork gradients that can be inserted while satisfying condition 16. Interpolated gradients $\widehat{\phi_i \phi_{\neg i}} = (\theta_t - \theta_{t-1}) \odot (\bigcap_i^N m_i)$ are gradients with different combinations of subnetwork masks and subnetwork values assigned in $m_i$ and $\theta_t$ accordingly; endpoint $\phi_i = \widehat{\phi_i \phi_{\neg i}} = (\theta_t - \theta_{t-1}) \odot m_i$. For our analysis of the multi-agent backdoor attack with respect to joint distribution shift, the gradient $\phi$ is a *subnetwork gradient* corresponding to a specific shift $\varepsilon \mapsto \phi$ (e.g. backdoor trigger pattern, or sub-population shift in clean inputs, or stylization): $. \theta_t = \theta_{t-1} + \sum_i^{|\{\phi\}|} \phi_i$.

### A.1.3 PRELIMINARIES ON JOINT DISTRIBUTION SHIFT

Distribution shifts can vary by *source* of distribution (e.g. domain shift, task shift, label shift) and *variations* per source (e.g. multiple backdoor triggers, multiple domains). *Joint distribution shift* is denoted as the phenomenon when distribution shift is attributed to multiple sources and/or variations per source. Eqt 8 is an example of how the multi-agent backdoor attack (multiple variations of backdoor attack) alters the probability density functions per label. To address joint distribution shift, $\phi$ should be transferable across a set of $\{\varepsilon\}$. One approach to inspecting this is by inspecting the insertion of subnetworks.

There is growing literature on the study of joint distribution shift. Naseer et al. (2019); Datta (2021); Qi et al. (2021a) show worsened model performance after applying adversarial perturbations upon domain-shifted inputs. Ganin et al. (2016) proposed a domain-adapted adversarial training scheme to improve domain adaptation performance. Geirhos et al. (2019) also show that the use of stylized perturbations with AdaIN as an augmentation procedure can improve performance on an adversarial perturbation dataset ImageNet-C. AdvTrojan (Liu et al., 2021) combines adversarial perturbations together with backdoor trigger perturbations to craft stealthy triggers to perform backdoor attacks. Weng et al. (2020) studies the trade-off between adversarial defenses optimized towards adversarial perturbations against backdoor defenses optimized towards backdoor perturbations. Santurkar et al. (2020) synthesize distribution shifts by combining random noise, adversarial perturbations, and domain shifts to varying levels to contribute subpopulation shift benchmarks. Rusak et al. (2020) proposed a robustness measure by augmenting a dataset with both adversarial noise and stylized perturbations, by evaluating a set of perturbation types including Gaussian noise, stylization and adversarial perturbations.

### A.1.4 BACKFIRING EFFECT: CHANGES IN DISTRIBUTION OF $\varepsilon$

**Lemma 1.** *For an input variable $\mathbf{X}(\omega)$ that is sampled randomly, the output variable $X(\omega)$ from operations $\varepsilon$ applied to $\mathbf{X}(\omega)$ will also tend to be random.*

*Proof.* A random variable $\mathbf{X}$ is a mapping from $W$ to $\mathbb{R}$, that is $\mathbf{X}(\omega) \in \mathbb{R}$ for $\omega \in \mathbb{R}$. $X(\omega) = \mathbf{X}(\omega) + \varepsilon$, thus $X$ is also a mapping $X : W \mapsto \mathbb{R}$. The measure for random variable $\mathbf{X}$ is defined by the cumulative distribution function $F(x) = \mathbf{P}(X \leq x)$. For $x > 0$, $F_X(x) = \mathbf{P}(X \leq x) = \mathbf{P}(\mathbf{X} + \varepsilon \leq x) = \mathbf{P}(\mathbf{X} \leq x - \varepsilon) = F_{\mathbf{X}}(x - \varepsilon)$. Thus $X(\omega)$ is also measurable and is a random variable defined on the sample space $W$.

**Lemma 2.** *Suppose a given model $f(x, y; \theta) = \theta \cdot x$ and loss $\mathcal{L}(x, y; \theta) = f(x) - y$. Suppose we sample backdoored observations $(x_i = \mathbf{x} + \varepsilon_i), (y \to y_i) \sim D_i$. The change in loss between clean to perturbed input is $\frac{\partial \mathcal{L}}{\partial \theta} = \varepsilon(\theta) + c$.*

*Proof.*

$$\Delta \mathcal{L} = \mathcal{L}(x_i, y_i; \theta) - \mathcal{L}(\mathbf{x}, y; \theta)$$
$$= [f(x_i, y_i; \theta) - f(\mathbf{x}, y; \theta)] - [y_i - y]$$
$$= \theta[x_i - \mathbf{x}] - [y_i - y]$$
$$\frac{\partial^2 \mathcal{L}}{\partial \theta^2} = x_i - \mathbf{x} = \varepsilon$$
$$\frac{\partial \mathcal{L}}{\partial \theta} = \varepsilon(\theta) + c$$

$\square$

**Theorem 1.** *Let $x, y \sim \mathbb{D} \setminus (D_0 \cup \{D_i\}^N)$ and $(\mathbf{x} + \varepsilon_{\text{noise}} + \{\varepsilon_i\}^N), (y \to y_i) \sim D_i$ be sampled clean and backdoored observations from their respective distributions. Let* Rand $: s \sim \mathcal{U}(S)$ *s.t.* $\mathbb{P}(s) = \frac{1}{|S|}$ *denote a random distribution where an observation $s$ is uniformly sampled from (discrete) set $S$. If $N \to \infty$, then it follows that predicted label $y^* = f(\mathbf{x} + \varepsilon; \theta) \sim \mathcal{U}(\mathcal{Y})$ s.t. $\mathbf{P}(y^*) = \frac{1}{|\mathcal{Y}|}$.*

**Proof sketch of Theorem 1.** With multiple attackers, we sample clean observations $x, y \sim \mathbb{D} \setminus (D_0 \cup \{D_i\}^N)$, backdoored observations $(\mathbf{x} + \varepsilon_{\text{noise}} + \{\varepsilon_i\}^N), (y \to y_i) \sim D_i$.

$$x = \mathbf{x} + \varepsilon$$
$$\mathcal{L}(x, y) = \mathcal{L}(\mathbf{x}, y) + \mathcal{L}(\varepsilon, y)$$
$$\frac{\partial \mathcal{L}(x, y)}{\partial \theta} = \frac{\partial \mathcal{L}(\mathbf{x}, y)}{\partial \theta} + \frac{\partial \mathcal{L}(\varepsilon, y)}{\partial \theta}$$
$$\Rightarrow \theta_t := \theta_{t-1} - \sum_{\mathbf{x}, y}^{X, Y} \frac{\partial \mathcal{L}(\mathbf{x}, y)}{\partial \theta} - \sum_{\varepsilon, y}^{X, Y} \frac{\partial \mathcal{L}(\varepsilon, y)}{\partial \theta}$$

This decomposition implies $\frac{\partial \mathcal{L}(\mathbf{x}, y)}{\partial \theta}$ updates part of $\theta$ w.r.t. $\mathbf{x}$, which we denote as $\theta \odot m_{\mathbf{x}}$, and $\frac{\partial \mathcal{L}(\varepsilon, y)}{\partial \theta}$ updates part of $\theta$ w.r.t. $\varepsilon$, which we denote as $\theta \odot m_\varepsilon$, where $m_{\mathbf{x}}, m_\varepsilon \in \{0, 1\}^{|\theta|}$ are masks of $\theta \equiv \theta \odot (m_{\mathbf{x}} + m_\varepsilon)$. Given the distances (squared Euclidean norm) between the shifted inputs and outputs $\mathbf{x} \to x_i$ and $y \to y_i$, we can enumerate the following 4 cases. Case (1) is approximately a single-agent backdoor attack, and is not evaluated. Cases (2)-(4) are variations of shifts in inputs and labels in a backdoor attack and manifest in our experiments.

$$\begin{cases} ||x_i - \mathbf{x}||_2^2 \approx 0 & , & ||y_i - y||_2^2 \approx 0 & \text{(Case 1)} \\ ||x_i - \mathbf{x}||_2^2 > 0 & , & ||y_i - y||_2^2 \approx 0 & \text{(Case 2)} \\ ||x_i - \mathbf{x}||_2^2 \approx 0 & , & ||y_i - y||_2^2 > 0 & \text{(Case 3)} \\ ||x_i - \mathbf{x}||_2^2 > 0 & , & ||y_i - y||_2^2 > 0 & \text{(Case 4)} \end{cases}$$

For $\varepsilon = \{\varepsilon_i\}^{i \in N+1}$, if $N \to \infty$, then $\varepsilon \sim$ Rand. We denote a random distribution Rand $: s \sim \mathcal{U}(S)$ s.t. $\mathbb{P}(s) = \frac{1}{|S|}$, where an observation $s$ is uniformly sampled from (discrete) set $S$. By Lemma 1 and 2, if $\varepsilon \sim$ Rand, then $\frac{\partial \mathcal{L}(\varepsilon, y; \theta)}{\partial \theta} \sim$ Rand and $f(x_i; \theta) - f(\mathbf{x}; \theta) \approx f(\varepsilon; \theta) \sim$ Rand.

Hence, for each case of $\frac{\partial \mathcal{L}(\varepsilon, y; \theta)}{\partial \theta}$:

If $\frac{\partial \mathcal{L}(\varepsilon, y; \theta)}{\partial \theta} \neq 0$, given $\theta = \theta \odot (m_{\mathbf{x}} + m_\varepsilon)$, then $f(\varepsilon; \theta) \approx f(\varepsilon; \theta + m_\varepsilon) \sim$ Rand;

If $\frac{\partial \mathcal{L}(\varepsilon, y; \theta)}{\partial \theta} = 0$, given $m_{\mathbf{x}} = 1^{|\theta|}, m_\varepsilon = 0^{|\theta|}$, then $f(\varepsilon; \theta) \approx f(\varepsilon; \theta + m_{\mathbf{x}}) \sim$ Rand.

In both cases, the predicted value of $f$ will be sampled randomly. Given it randomly samples from the label space $\mathcal{Y}$, in a multi-agent backdoor attack, and shifted input:output Cases (2)-(4), it follows that under the presence of a backdoor trigger pattern a prediction $y \sim \mathcal{U}(\mathcal{Y})$ s.t. $\mathbb{P}(y) = \frac{1}{|\mathcal{Y}|}$. The lower bound of attack success rate would be $\frac{1}{|\mathcal{Y}|}$ (0.1 for CIFAR-10).

### A.1.5 INSPECTING SUBNETWORK GRADIENTS: CHANGES IN PROBABILITY DISTRIBUTIONS W.R.T. $\mathcal{X}, \mathcal{Y}$-SPACE

**Theorem 3.** *Let $x, y \sim \mathbb{D} \setminus (D_0 \cup \{D_i\}^N)$ and $(\mathbf{x} + \varepsilon_{\text{noise}} + \{\varepsilon_i\}^N), (y \to y_i) \sim D_i$ be sampled clean and backdoored observations from their respective distributions. $\mathbf{P}_{x \to y}(x)$ denotes the probability density functions computing the likelihood that features of $x$ map to label $y$. A model $f$ can be approximated by $\mathbf{P}$ of all labels (Eqt 8). For any given pair of attacker indices $(i, \neg i)$ and their corresponding backdoor trigger patterns $(\varepsilon_i, \varepsilon_{\neg i})$ and target poison labels $(y_i, y_{\neg i})$, we formulate the updated model $f$ that can be approximated by $\mathbf{P}$ of all labels as Eqt 8. By analysis of cases and empirical results, the final prediction $f(x)$ is skewed w.r.t. the distribution of $\{\varepsilon\}$.*

**Proof sketch of Theorem 3.** Inductively demonstrated with different attack scenarios, we show that the model as a function approximator is composed of multiple probability density functions corresponding to each backdoor mapping $\varepsilon_i : y_i$.

***No Attack (N=0).*** We sample a set of clean observations $x, y \sim \mathbb{D}$. $\mathbf{P}_{x \to y}(x)$ denotes the probability density functions computing the likelihood that features of $x$ map to label $y$. A model $f$ can be

approximated by $\mathbf{P}$ of all labels $\mathbf{P}(x) = \{\mathbf{P}_{\mathbf{x}\to y}(\mathbf{x}) \cdot \mathbf{P}_{\varepsilon_{\mathrm{noise}}\to y}(\varepsilon_{\mathrm{noise}})\}^{y\in\mathcal{Y}}$, i.e.:

$$f(x;\theta) = \arg\max_{y\in\mathcal{Y}}\{\mathbf{P}_{\mathbf{x}\to y}(\mathbf{x}) \cdot \mathbf{P}_{\varepsilon_{\mathrm{noise}}\to y}(\varepsilon_{\mathrm{noise}})\} \tag{5}$$

***Single-Agent Backdoor Attack (N=1).*** We sample clean observations $x, y \sim \mathbb{D} \setminus D_0$ and backdoored observations $(\mathbf{x} + \varepsilon_{\mathrm{noise}} + \varepsilon_0), (y \to y_0) \sim D_0$, where $\varepsilon_0 > 0$ and $y \neq y_0$. Sampling an input from the joint distribution $x \sim \mathbb{D}$ where $\mathbb{D} = D_0 \cup (\mathbb{D} \setminus D_0)$, $x$ would be evaluated by $f$ with respect to all features (including perturbation feature). The newly-added perturbation feature $\varepsilon_0$ is evaluated by $f$, where it manifests in a given input or not (returns 0 if not), and requires a corresponding subnetwork gradient $\phi_0$. The proposed subnetwork gradient insertion $\phi_0$ is accepted if Eqt 15 is satisfied.

$$f(x;\theta + \phi_0) = \arg\max_{y\in\mathcal{Y}}\{\mathbf{P}_{\mathbf{x}\to y}(\mathbf{x}) \cdot \mathbf{P}_{\varepsilon_{\mathrm{noise}}\to y}(\varepsilon_{\mathrm{noise}}) \cdot \mathbf{P}_{\varepsilon_0\to y}(\varepsilon_0)\} \tag{6}$$

***Multi-Agent Backdoor Attack (N=2).*** We sample clean observations $x, y \sim \mathbb{D} \setminus (D_0 \cup D_1)$, backdoored observations $(\mathbf{x} + \varepsilon_{\mathrm{noise}} + \varepsilon_0), (y \to y_0) \sim D_0$ and $(\mathbf{x} + \varepsilon_{\mathrm{noise}} + \varepsilon_1), (y \to y_1) \sim D_1$, where $\varepsilon_0, \varepsilon_1 > 0$ and $y \neq y_0, y_1$.

There are 2 primary considerations to evaluate: (I) transfer/interference between features and labels between $D_0$ and $D_1$; and (II) loss reduction w.r.t. gradient selection. (I) manifests case-by-case, depending if in a particular case whether $||\varepsilon_0 - \varepsilon_1||_2^2 > 0$ or $||\varepsilon_0 - \varepsilon_1||_2^2 \approx 0$, whether $y_0 = y_1$ or $y_0 \neq y_1$. In terms of gradient selection, since there are at least 4 subnetwork gradient scenarios to evaluate: (i) no subnetwork gradient $[\theta]$, (ii) subnetwork gradient of $\varepsilon_0$ (endpoint) $[\theta + \phi_0]$, (iii) subnetwork gradient of $\varepsilon_1$ (endpoint) $[\theta + \phi_1]$, and (iv) interpolated subnetwork gradient between $\varepsilon_0$ and $\varepsilon_1$ $[\theta + \widehat{\phi_0\phi_1}]$. Sampling $x \sim \mathbb{D}$, each of these $\theta + \phi$ are evaluated case-by-case in Eqt 7. Among these candidate subnetwork gradients, the inserted (combination) of subnetwork gradients is determined by Eqt 17.

$$f(x;\theta + \phi) = \arg\max_{y\in\mathcal{Y}}\{\mathbf{P}_{\mathbf{x}\to y}(\mathbf{x}) \cdot \mathbf{P}_{\varepsilon_{\mathrm{noise}}\to y}(\varepsilon_{\mathrm{noise}}) \cdot \mathbf{P}_{\varepsilon_0\to y}(\varepsilon_0) \cdot \mathbf{P}_{\varepsilon_1\to y}(\varepsilon_1)\} \tag{7}$$

***Multi-Agent Backdoor Attack (N>1).*** Extending on our study of the 2-Attacker scenario, for any given pair of attacker indices $(i, \neg i)$, we need to consider the distances (squared Euclidean norm) of $(\varepsilon_i, \varepsilon_{\neg i})$ and $(y_i, y_{\neg i})$. By induction, we obtain Eqt 8, where $\phi$ is an interpolation of $N$ subnetworks to varying extents.

$$f(x;\theta + \phi) = \arg\max_{y\in\mathcal{Y}}\left\{\mathbf{P}_{\mathbf{x}\to y}(\mathbf{x}) \cdot \mathbf{P}_{\varepsilon_{\mathrm{noise}}\to y}(\varepsilon_{\mathrm{noise}}) \cdot \prod_i^N \mathbf{P}_{\varepsilon_i\to y}(\varepsilon_i)\right\} \tag{8}$$

We enumerate cases from Eqt 8, mapped similar to Theorem 1 cases. Note these are non-identical case mappings: Theorem 1 cases are evaluating distances between the unshifted and shifted inputs and labels in the joint dataset; Theorem 3 cases are evaluating distances between inputs and labels of private datasets of different attackers.

*(Case 1)* If $||\varepsilon_i - \varepsilon_{\neg i}||_2^2 \approx 0$ and $y_i = y_{\neg i}$, attackers approximate a single attacker $\{\varepsilon_0, y_0\}$, hence the collective attack success rate should approximate that of a single-agent backdoor attack.

*(Case 3)* If $y_i \neq y_{\neg i}$ and $||\varepsilon_i - \varepsilon_{\neg i}||_2^2 \approx 0$, then the feature collisions arising due to this label shift will cause conflicting label predictions from each $\mathbf{P}_{\varepsilon_i\to y}(\varepsilon_i)$ in Eqt 8, which will skew the final label prediction.

This manifests in escalation, where in $\boxed{\mathrm{E4}}$ we observe that if $|\{\varepsilon_i \to y_i\} - |\{\varepsilon_{\neg i} \to y_{\neg i}\}|$, then the attack success rate of $a_i$ would be better than $a_{\neg i}$. This manifests when there are a large number of attackers $|\varepsilon|$, where in $\boxed{\mathrm{E4}}$ we observe that many attackers with low distance perturbations but randomly-assigned target trigger labels tend to result in low collective attack success rate. This phenomenon may arise due to the model returning random label predictions during test-time if provided random labels during train-time, in-line with Theorem 1, and extending upon Zhang et al. (2017).

*(Cases 2 & 4)* If $||\varepsilon_i - \varepsilon_{\neg i}||_2^2 > 0$, whether $y_i = y_{\neg i}$ or $y_i \neq y_{\neg i}$, given the backdoor trigger patterns are distant in the feature space (minimal feature collision), it follows that the collective attack success rate should be more dependent on model capacity to store a unique subnetwork for each $\varepsilon$.

Empirically, this is neither in-line with respect to capacity findings in $\boxed{\mathrm{E2}}$ nor in-line with trigger distance findings in $\boxed{\mathrm{E4}}$. This informs us that, although the cosine distance indicates a great distance

between trigger patterns, feature collisions still occur in practice when $||\varepsilon_i - \varepsilon_{\neg i}||_2^2 > 0$. It indicates that *Case 3* (skewed label prediction) is more dominant in practice, and this is in-line with E4 where the cosine distance between trigger patterns are high, but $y_i = y_{\neg i}$ returns higher collective attack success rate than when $y_i \neq y_{\neg i}$.

### A.1.6 Inspecting subnetwork gradients: Changes in loss terms

**Theorem 2.** *A model of fixed capicity permits $\theta$ with limited subnetworks. Loss optimization condition (Eqt 16) constrains the insertion of subnetwork gradients $\phi$ to minimize total loss over the joint dataset. To satisfy the $\phi$-insertion condition* LHS < RHS *(16), other than imbalancing the loss terms with high poison rate (Lemma 3), Eqt 17 shows how the transferability of $\varepsilon$ determines whether its subnetwork gradient $\phi$ is accepted given $\varepsilon \mapsto \phi$. It is empirically demonstrated $|\{\varepsilon : \phi\}^*| \ll N$.*

**Proof sketch of Theorem 2.** Inductively demonstrated with different attack scenarios, we show how the loss function evaluates the insertion of a subnetwork w.r.t. its gradients. Pursuing a loss perspective on this problem is motivated by implications from the transfer-interference tradeoff (Riemer et al., 2019) on feature transferability, by implications from imbalanced gradients (Jiang et al., 2021) on how loss terms can overpower optimization pathways, and by implications of transfer loss as an implicit distance metric.

***Single-Agent Backdoor Attack (N=1).*** We consider the loss minimization procedure at this iteration as an implicit measurement of the entropy of the backdoor subnetwork; if there is marginal information:capacity benefit from the insertion of $\phi$ to $\theta$ compared to not inserting it, then the subnetwork gradient is added to $\theta$ in this iteration. As $\theta$ is already optimized to $\mathbb{D} \setminus D_i$, therefore $\frac{\partial \mathcal{L}(\mathbf{x},y)}{\partial \theta} \approx 0$ and thus resulting in Property 9. This update $\theta \to \theta^*$ is represented in Eqt 4, where the update condition $\mathcal{L}(\theta \cup \phi^{\text{backdoor}}) < \mathcal{L}(\theta)$ is defined by Eqt 10, consisting of loss with respect to both clean and poisoned inputs. We denote LHS (10) and RHS (10) as the left-hand side and right-hand side of an update condition (10) respectively. The subnetwork would be updated based on update condition (10), where the insertion of the subnetwork would be rejected if LHS > RHS (10). We refactor into Eqt (11) as an update condition: if LHS > RHS (11), then a subnetwork gradient insertion is rejected.

$$\frac{\partial \mathcal{L}(x,y)}{\partial \theta} = \frac{\partial \mathcal{L}(\mathbf{x},y)}{\partial \theta} + \frac{\partial \mathcal{L}(\varepsilon,y)}{\partial \theta} \approx \frac{\partial \mathcal{L}(\varepsilon,y)}{\partial \theta} \tag{9}$$

$$\mathcal{L}(\theta + \phi^{\text{backdoor}}; X^{\text{clean}}, Y^{\text{clean}}) + \mathcal{L}(\theta + \phi^{\text{backdoor}}; X^{\text{poison}}, Y^{\text{poison}}) < \mathcal{L}(\theta; X^{\text{clean}}, Y^{\text{clean}}) + \mathcal{L}(\theta; X^{\text{poison}}, Y^{\text{poison}}) \tag{10}$$

$$\mathcal{L}(\theta + \phi^{\text{backdoor}}; X^{\text{clean}}, Y^{\text{clean}}) - \mathcal{L}(\theta; X^{\text{clean}}, Y^{\text{clean}}) < \mathcal{L}(\theta; X^{\text{poison}}, Y^{\text{poison}}) - \mathcal{L}(\theta + \phi^{\text{backdoor}}; X^{\text{poison}}, Y^{\text{poison}}) \tag{11}$$

We refactor Eqt 11 into Eqt 14 after decomposing the backdoor inputs into clean and backdoor trigger features. To reiterate, to insert a candidate subnetwork gradient $\phi$, the aforementioned conditions 10-14 would need to be satisfied. To satisfy these conditions, at least 2 approaches can be taken: *(Case 1)* maximize the poison and perturbation rate, or *(Case 2)* jointly minimize the loss with respect to both clean inputs and backdoored inputs after the subnetwork gradient is inserted. From Figure 2 ($N = 10^0$), we know empirically that this condition can be satisfied for single-agent attacks.

$$\sum_{x,y}^{\mathbb{D} \setminus D_0} \left[ \mathcal{L}(\theta + \phi^{\text{backdoor}}; x^{\text{clean}}, y^{\text{clean}}) - \mathcal{L}(\theta; x^{\text{clean}}, y^{\text{clean}}) \right] < \sum_{x,y}^{D_0} \left[ \mathcal{L}(\theta; x^{\text{poison}}, y^{\text{poison}}) - \mathcal{L}(\theta + \phi^{\text{backdoor}}; x^{\text{poison}}, y^{\text{poison}}) \right] \tag{12}$$

$$\sum_{x,y}^{\mathbb{D} \setminus D_0} \left[ \mathcal{L}(\theta + \phi^{\text{backdoor}}; x^{\text{clean}}, y^{\text{clean}}) - \mathcal{L}(\theta; x^{\text{clean}}, y^{\text{clean}}) \right] < \sum_{x,y}^{D_0} \frac{|\mathbf{x}|}{|x|} \left[ \mathcal{L}(\theta; x^{\text{poison}}, y^{\text{poison}}) - \mathcal{L}(\theta + \phi^{\text{backdoor}}; x^{\text{poison}}, y^{\text{poison}}) \right]$$
$$+ \sum_{x,y}^{D_0} \frac{|\varepsilon|}{|x|} \left[ \mathcal{L}(\theta; x^{\text{poison}}, y^{\text{poison}}) - \mathcal{L}(\theta + \phi^{\text{backdoor}}; x^{\text{poison}}, y^{\text{poison}}) \right] \tag{13}$$

$$\frac{|\varepsilon|}{|x|} \sum_{x,y}^{D_0} \left[ \mathcal{L}(\theta; x^{\text{poison}}, y^{\text{poison}}) - \mathcal{L}(\theta + \phi^{\text{backdoor}}; x^{\text{poison}}, y^{\text{poison}}) \right]$$
$$> \left[ |\mathbb{D} \setminus D_0| + \frac{|\mathbf{x}|}{|x|} |D_0| \right] \cdot \frac{1}{|\mathbb{D} \setminus D_0|} \sum_{x,y}^{\mathbb{D} \setminus D_0} \left[ \mathcal{L}(\theta + \phi^{\text{backdoor}}; x^{\text{clean}}, y^{\text{clean}}) - \mathcal{L}(\theta; x^{\text{clean}}, y^{\text{clean}}) \right] \tag{14}$$

*(Case 1)* To maximize the poison and perturbation rate $D_0, \varepsilon$ alone, while keeping the loss values constant, we find that the lower bound required to satisfy conditions 10-14 is $\begin{cases} \frac{|\varepsilon|}{|x|} \geq \frac{1}{2} \\ |D_0| > 2|\mathbb{D} \setminus D_0| \end{cases}$ (Lemma 3). Causing an imbalance between the loss function terms is in-line with analysis in imbalanced gradients (Jiang et al., 2021). Considering the number of poisoned samples affects the information:capacity ratio, if the exclusion of $\phi^{\text{backdoor}}$ results in complete misclassification of $X^{\text{poison}}$, then for the same capacity requirements, each backdoor subnetwork has a high information:capacity ratio and it is possible for $\phi^{\text{backdoor}}$ to be accepted.

*(Case 2)* Agnostic to substantial poison/perturbation rate increases (Case 1), the attacker can also aim to craft backdoor trigger patterns that share transferable features to clean features (e.g. backdoor trigger patterns generated with PGD (Turner et al., 2019)). Given that $\phi$ is crafted such that it minimizes loss w.r.t. backdoored features (i.e. $\max(\text{RHS} - \text{LHS})$ (11) or $\max(\text{LHS} - \text{RHS})$ (14)), in order for Eqt 14 to be satisfied, the candidate subnetwork gradient will be accepted if it simultaneously minimizes loss w.r.t. clean features (i.e. $\min(\text{LHS} - \text{RHS})$ (11) or $\min(\text{RHS} - \text{LHS})$ (14)). We represent this dual condition as $\begin{cases} \frac{\partial \mathcal{L}(\mathbf{x},y)}{\partial \theta} < 0 \\ \frac{\partial \mathcal{L}(\varepsilon,y)}{\partial \theta} < 0 \end{cases}$. With $\mathbf{sign}(x) = \begin{cases} +1 & \text{for } x > 0 \\ -1 & \text{for } x < 0 \end{cases}$, condition 15 constrains the gradients to be in the same direction and loss to decrease, and must be satisfied to accept the candidate subnetwork gradient.

$$\mathbf{sign}(\frac{\partial \mathcal{L}(\mathbf{x}, y; \theta + \phi)}{\partial \theta}) + \mathbf{sign}(\frac{\partial \mathcal{L}(\varepsilon, y; \theta + \phi)}{\partial \theta}) \equiv -2 \tag{15}$$

***Multi-Agent Backdoor Attack (N>1).*** The addition of each backdoor attackers results in a corresponding subnetwork gradient, formulating $\theta = \sum_c^{|\mathbb{D}|} \phi_c + \sum_n^N \phi_{b_n}$ for $N$ attackers. The bound in *(N=2: Case 1)* persists for $N > 1$; in this analysis, we extend on *(N=2: Case 2)*. The cumulative poison rate $p$ is composed of the poison rates for each attacker $p = \sum_n^N p_n = \sum_n^N \frac{|D_n|}{\mathbb{D}}$. Each individual poison rate is thus smaller than the sum of all poison rates, thus the number of backdoored inputs allocated per backdoor subnetwork is smaller. We also presume that any one backdoor poison rate is not greater than the clean dataset, i.e. $(1-p) > p_n \forall n \in N$. Note that $(1-p)$ includes not only the defender's clean contribution, but also clean inputs contributed by each attacker. Given the capacity limitations of a DNN, if the number of attackers $N$ is very large resulting in many candidate subnetworks, not all of them can be inserted into $\theta$. Given fixed model capacity, $\theta$ can only include a limited number of subnetworks, and this number depends on the extent each subnetwork carries information that can reduce loss for multiple backdoored sets of inputs (transferability). We can approximate this transferability by studying how the loss changes with respect to this subnetwork and 2 sets of inputs.

$$\sum_{x,y}^{\mathbb{D} \setminus D_i} \mathcal{L}(\theta + \sum \phi_n^{\text{backdoor}}; x^{\text{clean}}, y^{\text{clean}}) + \sum_i^N \sum_{x,y}^{D_i} \mathcal{L}(\theta + \sum \phi_n^{\text{backdoor}}; x^{\text{poison}}, y^{\text{poison}})$$
$$< \sum_{x,y}^{\mathbb{D} \setminus D_i} \mathcal{L}(\theta; x^{\text{clean}}, y^{\text{clean}}) + \sum_i^N \sum_{x,y}^{D_i} \mathcal{L}(\theta; x^{\text{poison}}, y^{\text{poison}}) \tag{16}$$

Compared to the single-agent attack, the information:capacity ratio per backdoor subnetwork is diluted. We can infer this from Eqt 16 (a multi-agent extension of Eqt 12), where $N$ subnetworks are required to carry information to compute correct predictions for all $N$ backdoored sets compared to 1 in single-attacker scenario. The loss optimization procedure (Eqt 16) determines the selection of subnetworks gradients that should be selected to minimize total loss over the joint dataset. It implicitly determines which backdoored private datasets to ignore with respect to loss optimization, which we reflect in Eqt 17.

Given capacity limitations, every combination of backdoor subnetwork gradient is evaluated against every pair of private dataset, and evaluate whether it simultaneously (1) reduces the total loss (Eqt 10), and (2) returns joint loss reduction with respect to any pair of sub-datasets (Eqt 16).

$$\{\varepsilon : \phi\}^* := \underset{\{\varepsilon:\phi\}}{\arg\min} -(1 + |\{\varepsilon\}|) \equiv \underset{\{\varepsilon:\phi\}}{\arg\min} \left[ \mathbf{sign}(\frac{\partial \mathcal{L}(\mathbf{x}, y; \theta + \phi)}{\partial \theta}) + \sum_{\varepsilon, \phi}^{\{\varepsilon \mapsto \phi\}} \mathbf{sign}(\frac{\partial \mathcal{L}(\varepsilon, y; \theta + \phi)}{\partial \theta}) \right] \tag{17}$$

We extend update condition (16) into a subnetwork gradient set optimization procedure (Eqt 17), where loss optimization computes a set of backdoor subnetwork gradients that can minimize the total loss over as many private datasets.

To make a backdoor subnetwork more salient with respect to procedure (17), an attacker could (i) increase their individual $p_n$ (Lemma 3), (ii) have similar/transferable backdoor patterns and target poison labels as other attackers (or any other form of cooperative behavior). We empirically show this in $\boxed{\text{E4}}$.

With respect to $\boxed{\text{E3}}$, adversarial perturbations work because they re-use existing subnetworks in $\theta$ (i.e. $\phi^{\text{clean}}$) without the need to insert a new one. Stylized perturbations can be decomposed into style and content features; the content features may have transferability against unstylized content features thus there may be no subsequent change to $\phi^{\text{clean}}$, though the insertion of a new $\phi^{\text{style}}$ faces a similar insertion obstacle as $\phi^{\text{backdoor}}$.

Backdoor subnetworks can have varying distances from each other (e.g. depending on how similar the backdoor trigger patterns and corresponding target poison labels are). Measuring the distance between subnetworks would be one way of testing whether a subnetwork carries transferable features for multiple private datasets, as at least in the backdoor setting each candidate subnetwork tends to be mapped to a specific private dataset. Based on $\boxed{\text{E6}}$, we observe that the parameters diverge as $N$ increase per layer, indicating the low likelihood that at scale a large number of random trigger patterns can share common transferable backdoors. In other words, this supports the notion that each subnetwork is relatively unique for each trigger pattern and share low transferability across a set of private datasets $||\phi_i(x) - \phi_j(x)||_2^2 > 0$.

### A.1.7 Bounds for poison-rate-driven subnetwork insertion

**Lemma 3.** To satisfy condition 14 through an increase in poison and perturbation rate alone, assuming the ratio of the loss differences is 1 (i.e. there is a 1:1 tradeoff where the insertion or removal of the subnetwork will cause the same increase/decrease in loss), then the resulting lower bound is $\begin{cases} \frac{|\varepsilon|}{|x|} \geq \frac{1}{2} \\ |D_0| > 2|\mathbb{D} \setminus D_0| \end{cases}$ .

*Proof.* If $\frac{\mathcal{L}(\theta+\phi;X^{\text{backdoor}},Y^{\text{backdoor}})-\mathcal{L}(\theta;X^{\text{backdoor}},Y^{\text{backdoor}})}{\mathcal{L}(\theta+\phi;X^{\text{clean}},Y^{\text{clean}})-\mathcal{L}(\theta;X^{\text{clean}},Y^{\text{clean}})} = 1$ ,

$$\frac{|\varepsilon|}{|x|} > |\mathbb{D} \setminus D_0| + \frac{|\mathbf{x}|}{|x|}|D_0|$$

$$\frac{|\varepsilon|}{|x|} > |\mathbb{D} \setminus D_0| + 1 - \frac{|\varepsilon|}{|x|}|D_0|$$

$$(2\frac{|\varepsilon|}{|x|} - 1)|D_0| > |\mathbb{D} \setminus D_0|$$

For the last statement to be true, $2\frac{|\varepsilon|}{|x|} - 1$ must be positive:

$$2\frac{|\varepsilon|}{|x|} - 1 \geq 0$$

$$\frac{|\varepsilon|}{|x|} \geq \frac{1}{2}$$

To obtain the minimum poison rate $|D_0|$, we substitute the minimum perturbation rate $\frac{|\varepsilon|}{|x|} = \frac{1}{2}$ such that

$$|D_0| > 2|\mathbb{D} \setminus D_0|$$

$\square$

### A.1.8 Backdoor attack algorithm

**BadNet Gu et al. (2019a).** Within the given dimensions (length $l$ × width $w$ × channels $c$) of an input $x \in X$, a single backdoor trigger pattern $m$ replaces pixel values of $x$ inplace. Indices $(l, w, c)$

specify a specific pixel value in a matrix. $m$ is a mask of identical dimensions to $x$ that contains the perturbed pixel values, while $z$ is its corresponding binary mask of 1 at the location of a perturbation and 0 everywhere else, i.e.:

$$z(l, w, c) = \begin{cases} 1, & \text{if } m(l, w, c) > 0 \\ 0, & \text{if } m(l, w, c) = 0 \end{cases}$$

The trigger pattern be of any value, as long as it recurringly exists in a poisoned dataset mapped to a poisoned label. Examples include sparse and semantically-irrelevant perturbations (Eykholt et al., 2018; Guo et al., 2019), low-frequency semantic features (e.g. mask addition of accessories such as sunglasses (Wenger et al., 2021), and low-arching or narrow eyes (Stoica et al., 2017)). The poison rate is the proportion of the private dataset that is backdoored:

$$p = \frac{|X^{\text{poison}}|}{|X^{\text{clean}}| + |X^{\text{poison}}|}$$

$\odot$ being the element-wise product operator, the BadNet-generated backdoored input is:

$$x^{poison} = x \odot (1 - z) + m \odot z$$

$$b : X^{\text{poison}} := \{x \odot (1 - z) + m \odot z\}^{x \in X^{\text{poison}}}$$

**Random-BadNet.** We implement the baseline backdoor attack algorithm BadNet (Gu et al., 2019b) with the adaptation that, instead of a single square in the corner, we generate randomized pixels such that each attacker has their own specific trigger pattern (and avoid collisions). We verify these random trigger patterns as being functional for single-agent backdoor attacks at $N = 1$. Many existing backdoor implementations in literature, including the default BadNet implementation, propose a static trigger, such as a square in the corner of an image input. BadNet only requires a poison rate; we additionally introduce the perturbation rate $\varepsilon$, which determines how much of an image to perturb.

Extending on BadNet, $m_i$ is a randomly-generated trigger pattern, sampled per attacker $a_i$. We make use of seeded `numpy.random.choice`[1] and `numpy.random.uniform`[2] functions from the Python `numpy` library. Perturbation rate $\varepsilon_i$ dictates the likelihood that an index pixel $(l, w, c)$ will be perturbed, and is used to generate the shape mask. The actual perturbation value is randomly sampled. As the perturbation dimensions are not constrained, a higher $\varepsilon_i$ results in higher density of perturbations. We compute the shape mask $z_i$, perturbation mask $m_i$, and consequently random-trigger-generated backdoored input as follows:

$$z_i = \{\texttt{numpy.random.choice}([0, 1], \texttt{size} = l \times w, \texttt{p} = [1 - \varepsilon_i, \varepsilon_i]).\texttt{reshape}(l, w)\} \times c$$

$$m_i(l, w, c) = \begin{cases} \texttt{numpy.random.uniform}(0, 1) \times 255, & \\ & \text{if } z(l, w, c) = 1 \\ 0, & \text{if } z(l, w, c) = 0 \end{cases}$$

$$x_i^{poison} = x_i \odot (1 - z_i) + m_i \odot z_i$$

$$b : X_i^{\text{poison}} := \{x_i \odot (1 - z_i) + m_i \odot z_i\}^{x_i \in X_i^{\text{poison}}}$$

The distribution of target poison labels may or may not be random. The distribution of clean labels are random, as we randomly sample inputs from the attacker's private dataset to re-assign clean labels to target poison labels. As all our evaluation datasets have 10 classes, this means $\frac{1}{10}$ of all backdoored inputs have target poison labels equivalent to clean labels. We tabulate the raw accuracy w.r.t. poisoned labels; a more reflective attack success rate would be (`Acc w.r.t. poisoned labels` $- 0.1$).

**Orthogonal-BadNet.** We adapt Random-BadNet with orthogonality between $N$ backdoor trigger patterns. Orthogonal trigger patterns should retain high cosine distances, and be far apart from each other in the representation space. We optimize for maximizing cosine distance here for the reason that we suspect a possibility that the randomly-generated trigger patterns may in some cases incur feature collisions (Li et al., 2019), where we have 2 very similar features but tending towards 2 very different

---

[1]https://numpy.org/doc/stable/reference/random/generated/numpy.random.choice.html
[2]https://numpy.org/doc/stable/reference/random/generated/numpy.random.uniform.html

labels; hence, it may be in the interest of attackers to completely minimize this occurrence and generate distinctly different trigger patterns that occupy different regions of the representation space. One form of interpreting the intention of minimizing collisions between features (backdoor trigger patterns) is the intention of minimizing interference between these features; Cheung et al. (2019) introduced a method for continual learning where they would like to store a set of weights without inducing interference between them during training, and hence they generate a set of orthogonal context vectors that transforms the weights for each task such that each resulting matrix would reside in a very distant region of the representation space against each other. We adapt a similar implementation, but applying an orthogonal matrix that transforms the backdoor trigger patterns into residing in a distant region away from the other resultant trigger patterns.

First, we generate a base random trigger pattern, the source of information sharing and coordination between the $N$ trigger patterns (unlike Random-BadNet) In-line with Cheung et al. (2019), where we also use seeded `scipy.stats.ortho_group.rvs`[3] from the Python `scipy` library ($l = w$), we sample orthogonal matrices from the Haar distribution, multiply it against the original generated trigger pattern (clip values for colour range [0, 255]) to return an orthogonal/distant trigger pattern.

$$o_i = \{\texttt{scipy.stats.ortho\_group.rvs(l)}\} \times c$$

$$b : X_i^{\text{poison}} := \{x_i \odot (1 - z_i \odot o_i) + m_i \odot z_i \odot o_i\}^{x_i \in X_i^{\text{poison}}}$$

## A.2 EVALUATION DESIGN (EXTENDED)

### A.2.1 POISON RATE

The allocation of the joint dataset that each attacker is expected to contribute is assumed to be identical (only varying on the number of backdoored inputs); so the collective attacker allocation is $1 - V_d$, and the individual attacker allocation is $(1 - V_d) \times \frac{1}{N}$. Hence, the real poison rate is calculated as $\rho = (1 - V_d) \times \frac{1}{N} \times p$. We visualize the allocation breakdown in Figure 4.

We acknowledge that a decrease in number of attackers can result in more of the joint dataset available for poisoning, and this can result in a larger absolute number of poisoned samples if the poison rate stays constant. To counter this effect, we take into account the maximum number of attackers we wish to evaluate for an experiment, e.g. $N = 1000$, such that even as $N$ varies, the real poison rate per attacker stays constant.

### A.2.2 (E1) MULTI-AGENT ATTACK SUCCESS RATE (EXTENDED)

In this case, as we wish to test a large number of attackers $N = 1000$ with small poison rates $p = 0.1$ for completeness, we set the defender allocation to be small $V_d = 0.1$. This allocation gives sufficient space for 1000 attackers, and we also verify that this extremely poison rate can still manifest a, albeit weakened, backdoor attack at $N = 1$.

The traintime-runtime split of each attacker is 80-20% (80% of the attacker's private dataset is contributed to the joint dataset, 20% reserved for evaluating in run-time). The train-test split for the defender was 80-20% (80% of joint dataset used for training, 20% for validation). We trained a ResNet-18 (He et al., 2015) model with batch size 128 and with early stopping when loss converges (approximately 30 epochs, validation accuracy of $92 - 93\%$; loss convergence depends on pooled dataset structure and number of attackers). We use early-stopping for a large number of epochs, as this training scheme would be reused and ensures consistent loss convergence given varying training datasets (e.g. training a model on augmented dataset with backdoor adversarial training, training a model on stylized perturbations). We use a Stochastic Gradient Descent optimizer with 0.001 learning rate and 0.9 momentum, and cross entropy loss function.

We set the seed of pre-requisite libraries as 3407 for all procedures, except procedures that require each attacker to have distinctly different randomly-sampled values (e.g. trigger pattern generation) in which the seed value is the index of the attacker (starting from 0).

---

[3]https://docs.scipy.org/doc/scipy/reference/generated/scipy.stats.ortho_group.html

### A.2.3 (E2) GAME VARIATIONS (EXTENDED)

**Datasets** We provision 4 datasets, 2 being domain-adapted variants of the other 2. MNIST (10 classes, 60,000 inputs, 1 colour channel) (LeCun & Cortes, 2010) and SVNH (10 classes, 630,420 inputs, 3 colour channels) (Netzer et al., 2011) are a domain pair for digits. CIFAR10 (10 classes, 60,0000 inputs, 3 colour channels) (Krizhevsky, 2009) and STL10 (10 classes, 12,000 inputs, 3 colour channels) (Coates et al., 2011) are a domain pair for objects. We make use of the whole dataset instead of the pre-defined train-test splits provisioned, given that we would like to retain custom train-test splits for defenders and also because we have an additional run-time evaluation set for attackers (Figure 4).

**Capacity** We trained with the same training procedure as $\boxed{\text{E1}}$ (same splits, optimizers, loss functions) with the variation of the model architecture: SmallCNN (channels $[16, 32, 32]$) (Fort et al., 2020), ResNet-{9, 18, 34, 50, 101, 152} (He et al., 2015), Wide ResNet-{50, 101}-2 (Zagoruyko & Komodakis, 2016), VGG 11-layer model (with batch normalization) (Simonyan & Zisserman, 2015). Due to computational constraints, we wished to sample number of attackers $N$ for the following ranges $(1 \ldots 10, 10 \ldots 20, 20 \ldots 100, 100..500, 500 \ldots 1000)$, linearly-spaced these ranges into 3 segments, and evaluated on all the returned $N$. Other than ResNet, we included other architectures including VGG11 (a comparably large capacity model to Wide ResNet-101 in terms of number of parameters but with different architecture) and SmallCNN (a small capacity model).

### A.2.4 (E3) ADDITIONAL SHIFT SOURCES (EXTENDED)

**Adversarial perturbations.** For adversarial perturbations, we use the Fast Gradient Sign Method (FGSM) (Goodfellow et al., 2015). With this attack method, adversarial perturbation rate $\varepsilon_a = 0.1$ can sufficiently bring down the attack success rate comparable and similar to that of $\varepsilon_a = 1.0$; we do this to induce variance. Hence, we scale the perturbation against an upper limit $1.0$ in our experiments, i.e. $\varepsilon_a^{'} = 0.1 \times \varepsilon_a$. Adversarial perturbations are only introduced during test-time, and each attacker only crafts adversarial perturbations with respect to their own private dataset (i.e. they train their own surrogate models with the same training scheme as the defender, and do not have any access to the joint dataset to craft perturbations). It is also worth noting that FGSM computes perturbations with respect to the gradients of the attacker's surrogate model where this model was trained on the attacker's private dataset, which contains backdoored inputs mapped to poisoned labels, meaning the feature representation space is perturbed with respect to backdoor trigger patterns. We do not train a surrogate model with respect to the clean private dataset, as the intention of a surrogate model is to approximate the target defender's model which has been assumed to be poisoned, and it is also in the attacker's best interests to introduce adversarial perturbations even with respect to the backdoor perturbations, as long as a misclassification occurs (which we can verify with the clean-label accuracy).

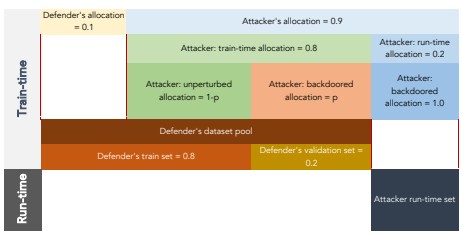

Figure 4: Summary of defender and attacker allocations in train-time and run-time. *Train-time* is the period where only training data is processed, including the defender's train set, defender's validation set, and attacker's train-time set. *Test-time* is the period where only test data is processed, being only attacker run-time set in this setup.

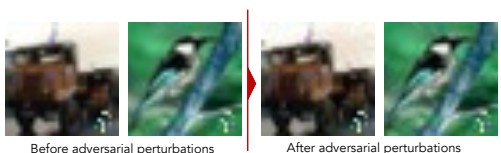

Before adversarial perturbations      After adversarial perturbations

Figure 5: Comparison of images before and after the insertion of adversarial perturbations. Backdoor trigger perturbations still exist, and the adversarial perturbations exist in other regions of the image.

**Stylized perturbations.** For stylistic perturbations, we use the Adaptive Instance Normalization (AdaIN) stylization method (Huang & Belongie, 2017), which is a standard method to stylize datasets such as stylized-ImageNet (Geirhos et al., 2019). Dataset stylization is considered as texture shift or domain shift in different literature. We randomly sample a distinct (non-repeating) style for each attacker. $\alpha$ is the degree of stylization to apply; 1.0 means 100% stylization, 0% means no stylization. We follow the implementation in Huang & Belongie (2017) and Geirhos et al. (2019) and stylize CIFAR-10 with the Paintings by Numbers style dataset. We adapt the method for our attack, by first randomly sampling a distinct set of styles for each attacker, and stylizing each attacker's sub-dataset before the insertion of backdoor or adversarial perturbations. This shift also contributes to the realistic scenario that different agents may have shifted datasets given heterogenous sources.

When the poison rate is 0.0, then the accuracy w.r.t. poisoned labels is equivalent to the accuracy w.r.t. clean labels. In our results, target poisoned labels are the intended labels based on attacker preferences, being clean labels for clean untriggered inputs and poisoned labels for backdoor triggered inputs. This means that when the run-time poison rate is 0.0, then the accuracy w.r.t. poison and clean labels are identical (unfiltered values in Figure 9). It also means that when $\varepsilon, p = 0.0$, then these values would be identical for accuracy w.r.t. poison as well as clean labels.

Prior to interpreting the results, we need to consider the attack objective of the attacker. For a backdoor attack, an attacker's objective is to maximize the accuracy w.r.t. poisoned labels. In an adversarial attack, an attacker's objective is to minimize the accuracy w.r.t. clean labels. The attack objectives may have additional conditions in literature, such as imperceptability to humans, or retaining high accuracy on clean inputs, etc, but the aforementioned 2 are the primary goals. They may not necessarily be contradictory either for two reasons: (i) if the poisoned label is non-identical to the clean label, then both a backdoor attack and adversarial attack will succeed in rendering a misclassification w.r.t. clean labels; and (ii) one similarity between a backdoor attack and adversarial attack is that they both rely on varying fidelity of information of the train-time distributions, where the backdoor attack has white-box knowledge of the perturbations that will cross the decision boundary to a target class, while the adversarial attack has grey/black-box knowledge of perturbations that may have a likelihood of crossing the decision boundary to a target class. In any case, we conclude for this evaluation that the attack objective of the attacker is to minimize the accuracy w.r.t. clean labels.

One of our suspicions regarding a low backdoor attack success rate is whether the generation of adversarial perturbations may possibly de-perturb backdoor perturbations: we visually inspect before-attack and after-attack images to verify that both adversarial and backdoor perturbations are retained in a multi-agent backdoor setting, and we sample a set in Figure 5.

With regards to $\varepsilon_\mathrm{b}, p = 1.0$ having such high ASR, it may not just be cause of a strong trigger pattern/saliency, but also it should be noted there is 100% class imbalance in this case for the attacker's surrogate model (only 1 class in the training set, hence no decision boundaries to cross; adversarial ASR should be 0.0). Given that our intention for this experiment is to observe the effect of a joint distribution shift between these two attacks unmodified in procedure and aligned as much as possible to their original attack design, we did not construct a coordinated adversarial-backdoor attack where only adversarial perturbations that do not counter, or even reinforce, the backdoor perturbations / poison labels are crafted.

A.2.5 (E4) COOPERATION OF AGENTS (EXTENDED)

In this setting, we primarily study 2 variables of cooperation, which are also the set of actions that an attacker can take: (i) input poison parameters ($p, \varepsilon$, and distance between different attackers' backdoor trigger patterns) , and (ii) target poison label selection. In addition to these 2 attacker actions that will formulate a set of strategies, we wish to evaluate the robustness of these strategies by (i) testing the scalability of the strategy at very large attacker counts, and (ii) testing the robustness of the strategy by introducing the weakest single-agent backdoor defense. We establish *information sharing* as the proce-

|  |  | $d$ | |
|---|---|---|---|
|  |  | *No Defense* | *Defense* |
|  | $\varepsilon = 0.55$ | $(26.2, 73.8)$ | $(13.7, 86.3)$ |
|  | 0% label overlap | $(8.6, 91.4)$ | $(12.3, 87.7)$ |
| $a$ | 100% label overlap | $(54.2, 45.8)$ | $(18.1, 81.9)$ |
|  | Random triggers | $(40.3, 59.7)$ | $(20.3, 79.7)$ |
|  | Orthogonal triggers | $(12.2, 87.8)$ | $(8.8, 91.2)$ |

Table 6: Expected values of each strategy from Tables 3 and 4, to determine that the approximate Nash Equilibrium is $(20.3, 79.7)$ when $a$ tends to use random triggers and $d$ uses a defense.

| Expt. | $V_d$ | train-test split (defender) | train-runtime split (attacker) | Model | Target poison label selection | $\varepsilon$ | $p$ | Real poison rate (#) |
|---|---|---|---|---|---|---|---|---|
| E1 | 0.1 | 80-20% | 80-20% | ResNet-18 | Random | - | 0.1, 0.2, 0.4, 0.8, 1.0 | 0.00009 (5), 0.00018 (11), 0.00036 (22), 0.00072 (44), 0.0009 (54) |
| E2 | 0.2 | 80-20% | 80-20% | ResNet-18 | Random | 0.55 | 0.55 | 0.0044 (264) |
| E3 | 0.1 | 80-20% | 80-20% | ResNet-18 | Random | - | 0.55 | 0.0005 (30) |
| E4 | 0.5 | 80-20% | 80-20% | ResNet-18 | - | 0.55 | 0.0, 0.2, 0.4, 0.6, 0.8, 1.0 | 0.0 (0), 0.001 (60), 0.002 (120), 0.003 (180), 0.004 (240), 0.005 (300) |
| E5 | 0.8 | 80-20% | 80-20% | ResNet-18 | Random | 0.55 | 0.55 | 0.001 (66) |
| E6 | 0.1 | 80-20% | 80-20% | SmallCNN | Random | 0.55 | 0.55 | 0.0005 (30) |

Table 5: Summary of default experimental configurations in each experiment. We also state the real poison rate (and corresponding number of train-time poisoned inputs). '-' indicates values that are varying by default.

dure in which agents send information between each other, and the collective information garnered can return outcomes in the range of anti-cooperative (which we denote as agents using information that hinders the other agent's individual payoff) to cooperative (which we denote as agents using information to maximize collective payoff). The payoff functions for anti-cooperative and non-cooperative strategies (i.e. individual ASR) are the same. We evaluate on 5 classes: 0 (airplane), 2 (bird), 4 (deer), 6 (frog), 8 (ship); we retain the same proportions of each of these 5 classes as $N$ varies. For $N = 100$, we specify the number of attackers $N\{Y\}$ that target class $Y$.

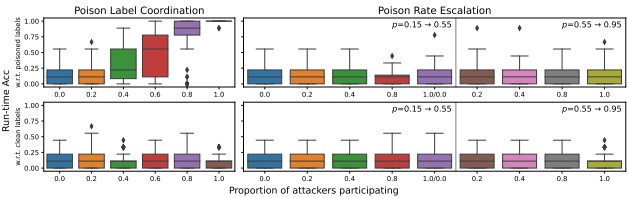

Figure 6: *Cooperation of agents ($N = 100$)*: (left) Coordination in terms of proportion of attackers selecting poison label 4, while others randomly select; (right) Escalation in terms of proportion of attackers increasing $p$ from 0.15 to 0.55, while others retain 0.15, then subsequently the increase in $p$ from 0.55 to 0.95, while others retain their previously-escalated 0.55; these escalation cases are plotted against corresponding distributions of accuracy w.r.t. poisoned labels (top row) and clean labels (bottom row).

With these strategies, we would like to observe the following agent dynamics-driven phenomenon, specifically the outcomes from attackers exercising various extents of selfishness (escalating attack parameters) against extents of collective goodwill (coordinating attack parameters): (i) the outcome from the escalation of $\varepsilon$; (ii) the outcome from a gradual coordination of target labels; (iii) the outcome of coordinating trigger pattern generation.

**Escalation.** Here, we describe how we demonstrate selfish escalation or collective coordination. While in other experiments, we scale the effect of random selection to a large number of attackers to approximate non-cooperative behaviour, we would now like to simulate simplified cases of anti-cooperative and cooperative strategies. For the selfish escalation of trigger patterns, we suppose each attacker crafts their backdoor trigger patterns independently from each other, and when information of the presence of other attackers is known or each attacker wishes to raise certainty of a backdoor attack, we consider the case where attackers to escalate their $\varepsilon$ from 0.15 to 0.55 to 0.95. To study trigger label collision cases and coordinated target label selection, we show a gradual change in trigger label selection amongst attackers, where they start off each having independent labels, then there is some trigger label collision between 40% of the attackers (40% attackers sharing the same label), then there is 100% trigger label collision between all of the attackers (i.e. all attackers share the same label). Coordination can manifest as either attackers each choosing distinctly different labels, or attackers all choosing the same label. To monitor collision between trigger patterns and trigger labels, we compute the cosine distance between each attacker's trigger pattern against that of Attacker Agent 1. The trigger patterns are randomly generated; first by computing a random set of pixel positional indices within image dimensions (pixel positions to be perturbed), which we

refer to as *shape* and show its corresponding cosine distance; then by computing the colour value change for each pixel position in the shape, which we refer this final trigger pattern of both positions and perturbations values as *shape+colour*. We assume there is no coordination hence the choice of random perturbations (as opposed to a perturbation-optimization function of minimizing cosine distance), though as $\varepsilon$ increases, we note that the cosine distance for both shape and shape+colour decrease (as the density of perturbations would be expected to be higher as $\varepsilon$ increases); this provides us with a range of trigger pattern distances in the representation space to evaluate against trigger label selection. In Table 3 and 4, for agents without the escalation in overlap of target label 4 (in red), we only redistribute the other 4 labels out equally, but in Figures 6 we redistribute 10 labels out randomly.

### A.2.6 (E5) PERFORMANCE AGAINST DEFENSES (EXTENDED)

We set the defender allocation to be significantly higher than that of other experiments because some of the defenses require subsets from the defender's private dataset to sample from, and enlarging this allows us to test the single-agent defenses leniently for the defenders and harshly for the attackers. This enlargement of defender allocation would also mean we should be careful when comparing values between experiments; for example, the real poison rate in this experiment is $0.0011(66)$, which has 12 more poisoned samples than $p = 1.0$ in E1, attributed to the large difference in $N$ considered.

Extending on stealth and imperceptability, an important aspect of the backdoor attack, there is a further sub-subclassification of backdoor attacks into *dirty*- (Gu et al., 2019a) and *clean*-label (Shafahi et al., 2018a; Zhu et al., 2019a) backdoor attacks. In dirty-label backdoor attacks, the true label of triggered inputs does not match the label assigned by the attacker (i.e., the sample would appear incorrectly labeled to a human). In clean-label backdoor attacks, the true label of triggered inputs matches the label assigned by the attacker. The 2 sub-classes can be executed with the same attack algorithm and follow the same underlying principles, with a change to the target trigger label, though variant algorithms for the clean-label algorithm also exist (Shafahi et al., 2018a; Zhu et al., 2019a).

**Clean Label Backdoor Attack.**    Hence, in addition to BadNet, we also evaluate defenses on the Clean Label Backdoor Attack (Turner et al., 2019). Also a common baseline backdoor attack algorithm, the main idea of their method is to perturb the poisoned samples such that the learning of salient characteristic of the input more difficult, hence causing the model to rely more heavily on the backdoor pattern in order to successfully perform label classification. It utilizes adversarial examples or GAN-generated data, such that the resulting poisoned inputs appear to be consistent with the clean labels and thus seem benign even upon human inspection. The objective of a targeted clean label poisoning attack (Shafahi et al., 2018a) (which also applies to a clean label backdoor attack), is to introduce backdoor perturbations to a set of inputs during train-time whose poisoned labels are equal to their original clean labels, but the usage of the backdoor pattern during run-time regardless of ground-truth class would return the poisoned label. We align our implementation with that of Turner et al. (2019), and use projected gradient descent (PGD) (Madry et al., 2018) to insert backdoor perturbations $\varepsilon$.

A lower accuracy w.r.t. poisoned labels infers a better defense. While a post-defense accuracy below 0.1 is indicative of mislabelling poisoned samples whose ground-truth clean labels were also poisoned labels, it is also at least indicative of de-salienating the backdoor trigger perturbation, and hence indicative of backdoor robustness.

**Defenses.**  We evaluate 2 augmentative (data augmentation, backdoor adversarial training) and 2 removal (spectral signatures, activation clustering) defenses. For augmentative defenses, $50\%$ of the defender's allocation of the dataset is assigned to augmentation: for $V_d = 0.8$, 0.4 is clean, 0.4 is augmented. We devised backdoor defenses based on the backfiring effect, such as agent augmentation and agent indexing, in Datta et al. (2021). To mitigate the low accuracy w.r.t. clean labels in the presence of a backdoor trigger, we devised a backdoor defense through the construction of compressed low-loss subspaces in Datta & Shadbolt (2022a).

- **No Defense:**  We retain identical defender model training conditions to that in E1. The defender allocation 0.8 would be unmodified during model training in this setting.
- **Data Augmentation:**   Recent evidence suggests that using strong data augmentation techniques (Borgnia et al., 2021) (e.g., CutMix (Yun et al., 2019) or MixUp (Zhang et al., 2018)) leads to a reduced backdoor attack success rate. We implement CutMix (Yun et al.,

2019), where augmentation takes place per batch, and training completes in accordance with aforementioned early stopping.

- **Backdoor Adversarial Training:** Geiping et al. (2021) extend the concept of adversarial training on defender-generated backdoor examples to insert their own triggers to existing labels. We implement backdoor adversarial training (Geiping et al., 2021), where the generation of backdoor perturbations is through BadNet (Gu et al., 2019a), where $50\%$ of the defender's allocation of the dataset is assigned to backdoor perturbation, $p, \varepsilon = 0.4$, and 20 different backdoor triggers used (i.e. allocation of defender's dataset for each backdoor trigger pattern is $(1 - 0.5) \times 0.8 \times \frac{1}{20}$ ).

- **Spectral Signatures:** Spectral Signatures (Tran et al., 2018) is an input inspection method used to perform subset removal from a training dataset. For each class in the backdoored dataset, the method uses the singular value decomposition of the covariance matix of the learned representation for each input in a class in order to compute an outlier score, and remove the top scores before re-training. In-line with existing implementations, we remove the top 5 scores for $N = 1$ attackers. For $N = 100$, we scale this value accordingly and remove the top 500 scores.

- **Activation Clustering:** Activation Clustering (Chen et al., 2018) is also an input inspection method used to perform subset removal from a training dataset. In-line with Chen et al. (2018)'s implementation, we perform dimensionality reduction using Independent Component Analysis (ICA) on the dataset activations, then use $k$-means clustering to separate the activations into two clusters, then use Exclusionary Reclassification to score and assess whether a given cluster corresponds to backdoored data and remove it.

### A.2.7 (E6) MODEL PARAMETERS INSPECTION (EXTENDED)

Other than $\varepsilon, p = 0.55$ and training SmallCNN, we retain the same attacker/defender configurations as E1. We adopt SmallCNN for its low parameter count, which would be helpful in simplifying the subnetwork generation and analysis, as well as overall interpretability. In addition, as we show that the backdoor attack is consistent across all model architectures (E2), using a smaller model means the subnetworks with respect to the complete network is less diluted and larger in proportion (and hence the cosine distance would be less diluted for our observation).

**Lottery ticket.** We start with a fixed random initialization, shared across the 4 models trained on $N = 1, 10, 100, 1000$. We use Frankle & Carbin (2019b)'s Iterative Magnitude Pruning (IMP) procedure to generate a pruned DNN (0.8% in size of the full DNN across all $N$s), also denoted as the lottery ticket. The lottery ticket is a subnetwork, specifically the set of nodes in the full DNN that are sufficient for inference at a similar performance as the unpruned DNN. The study of the lottery ticket helps us make some inferences with respect to how the feature space changes, as well as where the optima on the loss landscape has deviated. We compute the cosine distance per layer between (i) parameters in the full DNN, (ii) mask of the lottery ticket against the DNN, (iii) parameters of the lottery ticket. We plot the values of $N$ v.s. $N$ for the weights and biases matrices for each convolutional layer (conv2d{layer_index}) and fully-connected layer (fc{layer_index}). All in all, the lottery ticket is a proxy for the most salient features, hence also acts as an alternate feature space representation.

We compute the full network distance, as we wish to decompose the distance changes of the subnetwork across $N$ with respect to both new optima w.r.t. change in $N$ in addition to changes to the DNN w.r.t. the introduction of new trigger patterns. The mask is a one-zero positional matrix; rather than removing the zeroes, we compute the distance including zeros to retain the original dimensionality of the full network, and measure the distance with factoring in the position of the values. To retain the position of parameter values, we multiply the one-zero mask against the full network parameters. While the size (new parameters count) of the pruned network can vary, we set the threshold to stop pruning at 97%, where pruning stops after we maximize accuracy for 97% pruned weights. Specifically, the lottery tickets across all $N$s are 0.8% in size of the full DNN; in other words, we pruned the count from 15,722 to 126. Though we may expect that if a DNN were allowed to store as many subnetworks per triggers as possible if we let the pruning threshold to be variable, by setting the capacity to be fixed (in-line with our theoretical analysis), we let the optimization steps manifest the loss function tradeoff discussed earlier, and manifest the acceptance/rejection of backdoor subnetwork insertion for fixed lottery ticket generation.

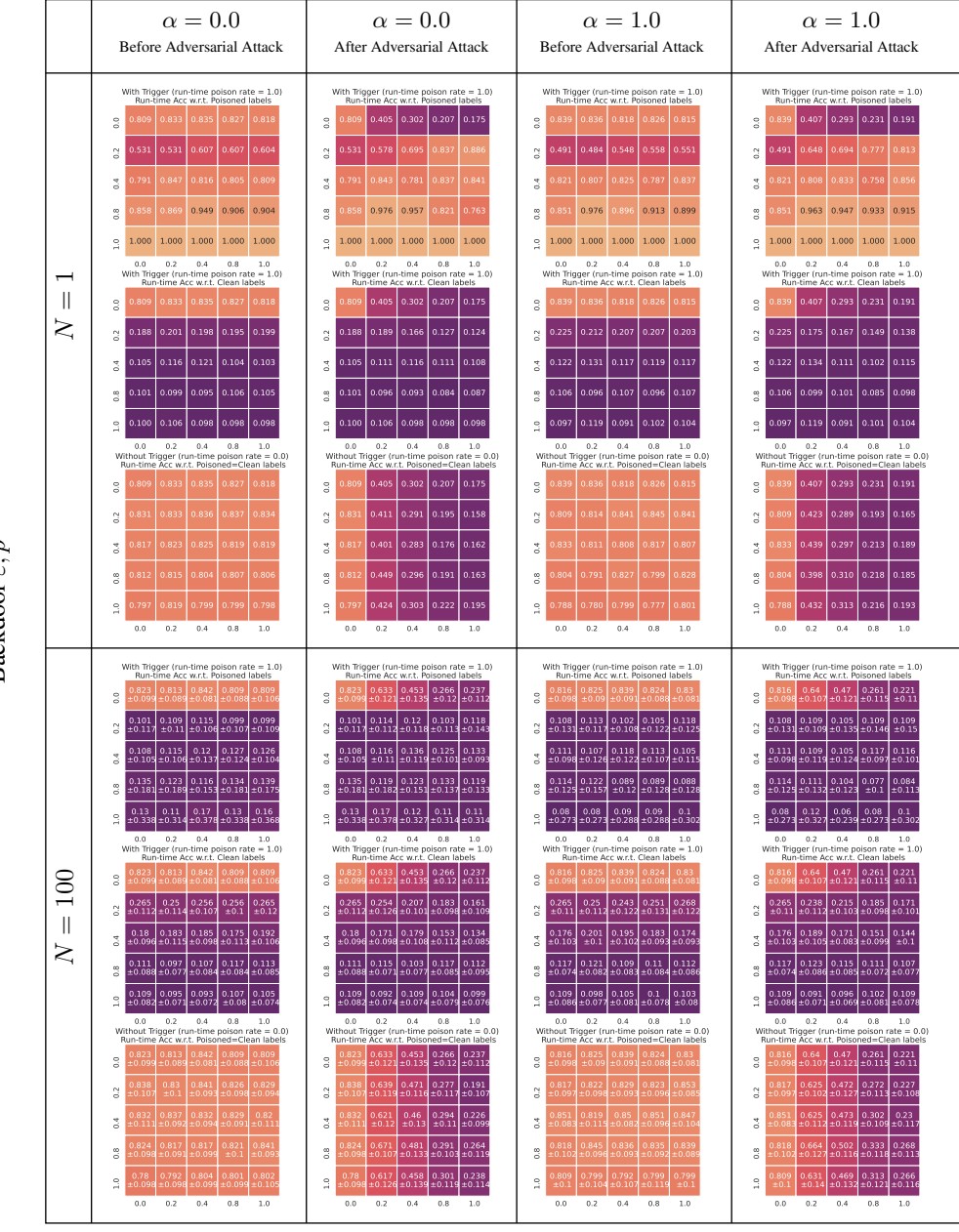

Figure 7: *Additional shift sources*: Joint distribution shift of varying counts: indexing columns from left to right, column 0 (Multiple {backdoor} perturbations; shifts=1), column 1 (Multiple {backdoor, adversarial} perturbations; shifts=2), column 2 (Multiple {backdoor, stylized} perturbations; shifts=2), column 3 (Multiple {backdoor, adversarial, stylized} perturbations; shifts=3).

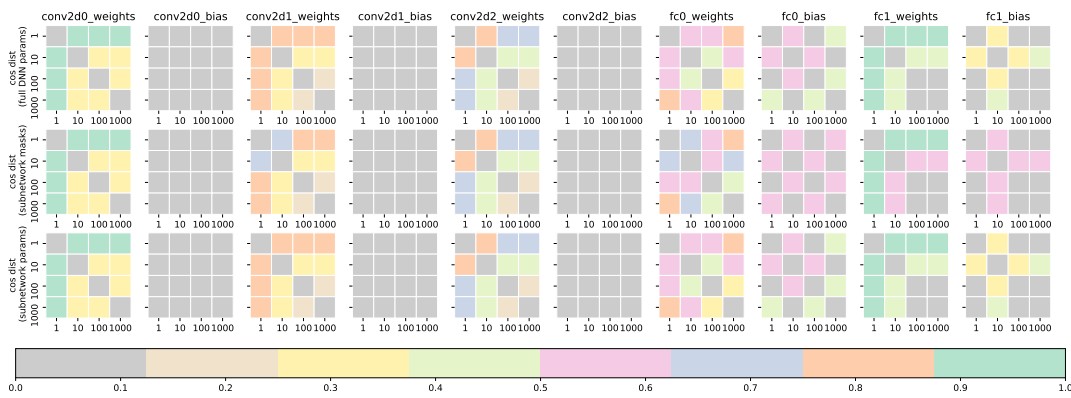

Figure 8: *Model parameters inspection*: Cosine distance of lottery ticket and original DNN per layer across $N$.

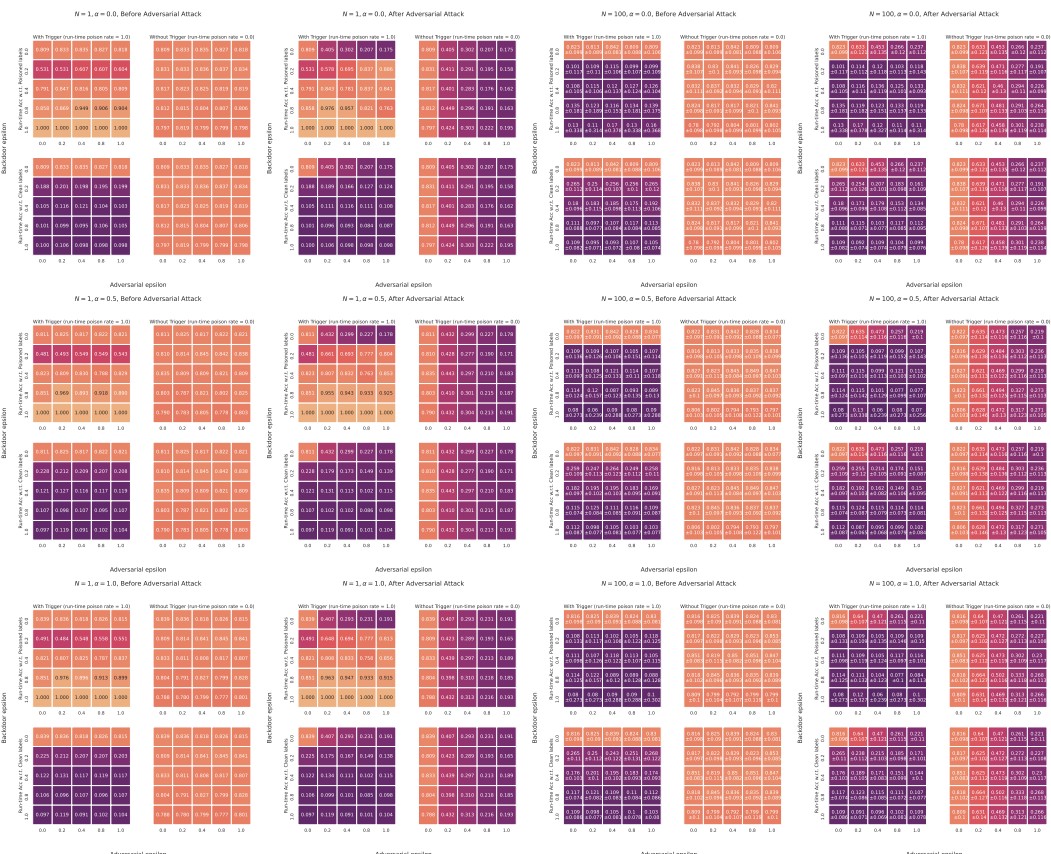

Figure 9: *Additional shift sources*: Unfiltered for $\alpha = 0, 0.5, 1.0$ and accuracy w.r.t. clean labels

