# OpenReview forum: "Backdoors Stuck At The Frontdoor: Multi-Agent Backdoor Attacks That Backfire"
_ICLR.cc/2023/Conference — Submitted to ICLR 2023_

### Official Review · Reviewer_eZzf · 2022-10-23

**Confidence:** 3
**Correctness:** 2
**Technical Novelty And Significance:** 2
**Empirical Novelty And Significance:** 2
**Recommendation:** 1

**Clarity, Quality, Novelty And Reproducibility:**

This paper might be novel, but not clear at all. The quality of writing is so bad.

**Strength And Weaknesses:**

I cannot find some strength of this paper. Below I write weaknesses.

- Unclear theoretical results
    - From the information provided in the main body, I could not get why Theorem 1 is true. Why does the attack success rate goes to 1/|Y| as the number of attackers go to infinity?
    - What is the key statement of Theorem 2? It seems like the setting & statement is shuffled in current version.
- Presentation of experimental results are not friendly to readers
    - Table 3 and 4 are containing many numbers, but what should we focus on? What is the take-away message?
    - Which text is for explaining Table 1 and 2? I was hard to get what is the take-away message of these results.
- Notation is not standard, some terms are defined unclearly
    - $x = \vx + \varepsilon$ is very weird. $x$ is usually for scalar, and $\vx$ is usually for vector.
    - What is “shifted X:Y distributions” in page 3?
    - $f(x;\theta)$ seems to be the output of a classifier parameterized by $\theta$, but not specified
    - accuracy expression is weird. If predicted label $f(x;\theta)$ is smaller than $y$, then it is considered as correct classification? And the maximum accuracy according to this expression is 2|X|, not 1.

**Summary Of The Paper:**

This paper focuses on multi-agent backdoor attack setup where multiple attackers inject backdoors for a single target model. The authors claim that this multi-agent attack can be defended by stable, natural defense.

**Summary Of The Review:**

Due to the weaknesses written above, I think it is not worth publishing.

---

### Official Review · Reviewer_JzAB · 2022-10-24

**Confidence:** 3
**Correctness:** 3
**Technical Novelty And Significance:** 3
**Empirical Novelty And Significance:** 3
**Recommendation:** 5

**Clarity, Quality, Novelty And Reproducibility:**

As described in the previous section, the text seems to be a bit lagging in terms of clarity. I find the scale of the experiments sufficient though, and having some value for the community. Further theoretical analysis would greatly improve this work’s novelty, which seems to be a bit limited otherwise. The implementation is based on a standard previous algorithm and the datasets are publicly available (by the way, why do the authors mention CIFAR at the beginning of section 4.1 when other datasets are apparently used as I gather from section E2), so the experiments may be reproducible, even though having the authors code at hand may definitely help, given the complexity of this setting.


**Strength And Weaknesses:**

I definitely agree that studying collaborative effects or interdependent behavioral dynamics arising in multi-agent interactions is an important direction which deserves to be explored, which holds true for the outsourced data collection problem studied in this paper as well. The authors present a substantial series of results, and the appendix is extensive. They also seem to make considerable effort to position themselves into the literature on the topic.

My main concern, though, related to the quality of this work’s exposition. Despite its clear potential, I feel like the authors are not doing the greatest job of explaining the entire setting, describing the involved concepts and pointing out the importance and range of scenarios their theoretical results apply to. The prose is often difficult to understand or even incoherent, and even after going over the text multiple times I still found myself confused at times. I wonder if this sentiment will be shared among the other reviewers too. For example, I believe the relation of the subnetwork gradients to the designed game needs to be explained in more detail, the main text lacks a description of which equilibrium it aims to attain, or if the players may face an equilibrium selection problem. Some of the problems are addressed in the appendix, but without it the reader is left guessing.

This apparent non-self-containment is further aggravated by referring the reader to Figures or Equations in the appendix, which are not supplementary (as some content understandably might be, e.g., the proofs), but rather required to understand the description of the results. I also fail to understand the motivation of the defender(s) in this game, who instead of focusing on their own model accuracy aim to minimize mean attacker rewards instead. Doesn’t that mean this may potentially substantially compromise the quality of their model? Another misleading thing about this work seems to be that despite introducing the game as a game between M defenders and N attackers, the authors are quick to restrict themselves to M=1.

I also remain confused about how to understand the takeaways listed in the conclusion. The authors claim that “existing models may not require significant defenses to block multi-agent backdoor attacks”, but at the same time, the second takeaway reads “the effectiveness of existing backdoor defenses drop when the number of attackers increase”. This appears to me to be rather contradictory.

There are also others, albein minor, things that suggest the text might have been rushed and need to be further polished. For example, the figures and tables are often too small which makes the reader repeatedly zoom in and out during reading, and sometimes they are even positioned tightly into the text which hampers reading as well. I would also suggest using some other color scheme (maybe somehow ‘logarithmic’) in Figure 2 such that the right two matrices are not all in the same color despite the values being different. The equations and the tables also overflow at times.


**Summary Of The Paper:**

This work studies a data-backdooring game in which multiple attackers strive to poison a training dataset of the defender(s) to affect classification on selected targets while preserving accuracy on the rest of the samples. After introducing related work, the authors formally describe the game the attackers and defender(s) are playing. The rest, comprising the main part of the text, is dedicated to empirical analysis of the game and description of the findings. According to the authors, the key takeaway is that in this setting, the attackers may experience huge drops in terms of their collective attack rates.

**Summary Of The Review:**

I like the idea of this work and the authors present an extensive amount of work, however, the text does not seem to be polished enough and may benefit from a better developed exposition.

---

### Official Review · Reviewer_qJJN · 2022-10-25

**Confidence:** 2
**Clarity, Quality, Novelty And Reproducibility:** See the above.
**Correctness:** 3
**Technical Novelty And Significance:** 2
**Empirical Novelty And Significance:** 2
**Recommendation:** 5

**Details Of Ethics Concerns:**

If the proposed attack is realistic, it may attack the current ML model.

**Strength And Weaknesses:**

This paper proposes a new scenario of the multi-agent backdoor attack and focuses on the cooperative dynamics between multiple attackers. They evaluate their proposed attack in experiments, but the fundamental difference between this cooperative attack and the existing work on a single attacker with multiple attacks is unclear. Some parts are hard to read, and the solution concept of the game is not well-defined.

What is the backdoor trigger?

Why do we need to consider the problem with multiple backdoor attackers? Are there any real-world scenarios?

The authors may need to give more details on multiple backdoor attackers in "the poisoning of crowdsourced and agent-driven datasets on Google Images (hence afflicting subsequent scraped datasets) and financial market data respectively, or poisoning through human-in-the-loop learning on mobile devices or social network platforms". It is not clear to me about the real-world problem with multiple backdoor attackers and why the model with a single attacker cannot handle it.

In my opinion, this paper is quite raw. Even though this paper handles a defender-attacker scenario, this paper does not mention any work on security games. It is unclear which solution concept (equilibrium) is used to describe the stable state of the interaction of these agents (the defender and the attackers). It is unclear if the proposed method can achieve this stable state. In Appendix, it seems that the authors use Nash equilibrium as the solution concept, then they need to justify why it is a Nash equilibrium instead of a Stackelberg equilibrium. In the defender-attacker scenario, the attacker usually can observe the defender strategy, and then the Stackelberg equilibrium is more suitable.

If this paper only considers the cooperation of attackers, then I cannot see the difference between this work and previous work with one defender and one attacker with multiple attacks.
If this paper considers the non-cooperation of attackers, it needs to discuss the solution concept among attackers, which is missing in the current paper.

Minor:
situations, It ->situations, it

**Summary Of The Paper:**

This paper investigates a multi-agent backdoor attack scenario, where multiple attackers attempt to backdoor a victim model simultaneously. Backdoor attacks, where an attacker poisons a model during training to successfully achieve targeted misclassification, are a major concern to train-time robustness. A consistent backfiring phenomenon is observed across a wide range of games, where agents suffer from a low collective attack success rate. They examine different modes of backdoor attack configurations.

**Summary Of The Review:**

This paper proposes a new scenario of the multi-agent backdoor attack and focuses on the cooperative dynamics between multiple attackers. They evaluate their proposed attack in experiments, but the fundamental difference between this cooperative attack and the existing work on a single attacker with multiple attacks is unclear. Some parts are hard to read, and the solution concept of the game is not well-defined.

---

### Official Review · Reviewer_q38C · 2022-10-26

**Confidence:** 4
**Clarity, Quality, Novelty And Reproducibility:** Novel and interesting paper.
**Correctness:** 4
**Technical Novelty And Significance:** 2
**Empirical Novelty And Significance:** 3
**Recommendation:** 3

**Strength And Weaknesses:**

Strength:

(1). The paper studied a novel and interesting topic in backdoor attack community, where multiple attackers exist in the environment and each attacker only performs backdoor poisoning on its private dataset.

Weaknesses:

(1). I am somewhat confused by the topic of the paper. Is the paper trying to demonstrate that if a training set is composed of multiple private subsets, and each subset can be backdoor attacked by an individual attacker, then the model learned jointly from the combined dataset may have low attack success rate? If so, I believe the discovery is not surprising and not super interesting to the community. The presentation of the paper needs to be improved significantly and the problem setup should be described more rigorously. Right now the section 3 is very confusing and some notations do not have a clear explanation. For example, the input space X=R^{l*w*c}. What is l, w, and c?

(2). I find the setting of the problem not super interesting. In particular, the paper considered an environment where each private subset can be attacked, and some defender is learning on a global dataset constructed by combing every poisoned subset. I am wondering in what real-world situations would such an environment exist? It's like the whole environment consists of attackers with different backdoor goals. A more reasonable situation is when only a small fraction of private subsets can be poisoned by an attacker, and most subsets are benign. In this case, one would be interested in designing defense mechanisms using clean data.

(3). The paper implies that a potential defense is to have multiple attackers. However, this is really not defense from the learner's perspective, because the leaner did not do anything at all to combat attackers in this case. Instead, the learner does nothing, but the attackers hurt each other. The paper does not propose any novel defense algorithms in this regard, which makes the paper very weak.

(4). The theoretical results need more intuitive explanation. I am wondering if the authors could explain the main messages that the authors want to convey with those results. Right now, the theory seems a bit disconnected to the main topic of the paper. For example, Theorem 1 suggests that as the number of attackers go to infinity, the prediction of the learned model becomes uniform. This result shows that the model performance decreases to the worst random predictor as the number of attackers grows. However, I don't see how this result connects to the backdoor attack studied in this paper. Why this result is interesting from the multi-attacker backdoor poisoning perspective. The authors may want to rethink their study and restate their main results to make them more coherent.

**Summary Of The Paper:**

This paper studied the effect of backdoor attacks in the multi-agent attacker scenario, where each attacker constructs backdoor attacks on  it's individual dataset. There is a single defender learner who receives poisoned datasets from all attackers, and learn a model on that dataset. The paper discovered a backfire effect, which is that when there are multiple attackers, the attack success rate drops. The paper introduced a set of cooperative dynamics between multiple attackers, extending on existing backdoor attack procedures. Experiments demonstrate the findings of the paper.

**Summary Of The Review:**

I work in related areas.

---

### Decision · Program_Chairs · 2023-01-20

**Decision:**

Reject

**Justification For Why Not Higher Score:**

There was a consensus among the reviewers that the work is not yet ready for publication.

**Justification For Why Not Lower Score:**

N/A

**Metareview: Summary, Strengths And Weaknesses:**

The reviewers agreed that the paper investigates a novel setting of backdoor attacks in multi-agent scenarios and that the problem setup is interesting. However, the reviewers pointed out several weaknesses in the paper, and there was a consensus that the work is not yet ready for publication. The reviewers have provided detailed and constructive feedback to the authors. We hope the authors can incorporate this feedback when preparing future revisions of the paper.